# CNS myelination requires VAMP2/3-mediated membrane expansion in oligodendrocytes

Mable Lam[1,4], Koji Takeo[1,3,4], Rafael G. Almeida[2], Madeline H. Cooper[1], Kathryn Wu[1], Manasi Iyer[1], Husniye Kantarci[1] & J. Bradley Zuchero [1]✉

Myelin is required for rapid nerve signaling and is emerging as a key driver of CNS plasticity and disease. How myelin is built and remodeled remains a fundamental question of neurobiology. Central to myelination is the ability of oligodendrocytes to add vast amounts of new cell membrane, expanding their surface areas by many thousand-fold. However, how oligodendrocytes add new membrane to build or remodel myelin is not fully understood. Here, we show that CNS myelin membrane addition requires exocytosis mediated by the vesicular SNARE proteins VAMP2/3. Genetic inactivation of VAMP2/3 in myelinating oligodendrocytes caused severe hypomyelination and premature death without overt loss of oligodendrocytes. Through live imaging, we discovered that VAMP2/3-mediated exocytosis drives membrane expansion within myelin sheaths to initiate wrapping and power sheath elongation. In conjunction with membrane expansion, mass spectrometry of oligodendrocyte surface proteins revealed that VAMP2/3 incorporates axon-myelin adhesion proteins that are collectively required to form nodes of Ranvier. Together, our results demonstrate that VAMP2/3-mediated membrane expansion in oligodendrocytes is indispensable for myelin formation, uncovering a cellular pathway that could sculpt myelination patterns in response to activity-dependent signals or be therapeutically targeted to promote regeneration in disease.

Myelinating oligodendrocytes spirally wrap and elongate numerous myelin sheaths on neuronal axons, tailoring myelin sheath thickness and length to modulate conduction velocity according to developmental and activity-dependent neuronal signals[1–3]. Each myelin sheath is a continuous extension of the oligodendrocyte plasma membrane, and a single oligodendrocyte has been estimated to increase its membrane area by several thousand-fold[4], requiring extensive addition of new membrane. Myelin sheaths are thought to grow in thickness and length by extending their innermost layer, or inner tongue, around and along the axon[5,6], but the cellular machinery that adds new membrane to power myelination is unknown. Beyond developmental myelination, experience and learning during adulthood can induce oligodendrocytes to form new myelin or restructure existing myelin sheaths, adapting myelination patterns to altered neural circuitry and potentially accelerating regeneration after demyelination[7–13]. This dynamic ability to make new myelin depends upon spatially coordinated membrane expansion. How oligodendrocytes add new membrane to build and remodel myelin sheaths remains an important, unanswered question in neurobiology.

[1]Department of Neurosurgery, Stanford University School of Medicine, Stanford, CA, USA. [2]Centre for Discovery Brain Sciences, University of Edinburgh, Edinburgh, UK. [3]Present address: Pharmaceutical Research Laboratories, Toray Industries, Inc., Kamakura, Kanagawa, Japan. [4]These authors contributed equally: Mable Lam, Koji Takeo. ✉e-mail: zuchero@stanford.edu

Oligodendrocytes may add new membrane through several mechanisms, including vesicular trafficking to the cell surface[14,15], non-vesicular lipid transport[16], or membrane incorporation of lipoproteins[17]. These membrane addition mechanisms have known roles in regulating cell signaling (e.g., synaptic vesicle release) and lipid metabolism, but less is known about their role in shaping cell morphology and size. Vesicular membrane addition occurs through membrane fusion by soluble *N*-ethylmaleimide sensitive factor attachment protein receptor (SNARE) complexes, where vesicular SNARE proteins (v-SNAREs) pair with their cognate SNARE partners on the target membrane to trigger exocytosis. In myelin, vesicles have been observed in the inner tongue and are thought to travel through "myelinic channels" extending through compact myelin regions[5,6,18–20], but the molecular components and function of these vesicles are unclear. Previous studies reported the upregulation of v-SNARE isoforms VAMP3 and VAMP7 in myelinating oligodendrocytes[21,22]. However, knockout of VAMP3 has no effect on myelination, and mislocalization of VAMP7 causes only mild developmental dysmyelination[22]. Thus, whether SNARE-mediated exocytosis is required for myelination remains unresolved.

More recently, RNA sequencing data revealed that *Vamp2* is expressed at levels comparable to *Vamp3* and *Vamp7* in oligodendrocytes, with little to no expression of other *Vamp* isoforms (Supplementary Fig. 1a, b)[23,24]. Although VAMP2 is best-characterized in synaptic vesicle release, recent studies demonstrate that VAMP2-mediated exocytosis is also necessary for membrane expansion of growing neurons in culture[25,26]. Since VAMP2 and VAMP3 are individually sufficient to drive vesicle fusion with the plasma membrane[27], compensation by VAMP2 may have limited the ability of previous VAMP3 KO studies to reveal the requirement of SNARE-mediated exocytosis in myelination. Therefore, we investigated the role of both v-SNARE isoforms in assembling myelin ultrastructure in nervous system development.

Here we demonstrate that VAMP2 and VAMP3, herein referred to as VAMP2/3, drive large-scale membrane expansion to power myelin sheath wrapping and elongation by directing vesicle fusion within myelin sheaths. VAMP2/3-associated vesicles coordinately deliver new membrane and axon-myelin adhesion proteins to the inner tongue and paranodes to assemble nodes of Ranvier. Thus, VAMP2/3-mediated exocytosis in oligodendrocytes represents a regulatory nexus for myelin sheath growth and node of Ranvier formation to sculpt neural circuitry during development.

## Results

### VAMP2/3-mediated exocytosis is required for CNS myelination
We first examined the expression of *Vamp2* and *Vamp3* across the oligodendrocyte lineage in the developing spinal cord white matter using multiplexed fluorescence RNA in situ hybridization (RNAscope) (Supplementary Fig. 1c, d). *Vamp2* was expressed at all differentiation stages of oligodendrocyte-lineage cells in vivo, while *Vamp3* was upregulated in mature oligodendrocytes marked by *Aspa* (Supplementary Fig. 1e, f). To address the potential redundant functions of VAMP2 and VAMP3 targeting to the plasma membrane, we used a transgenic mouse line ("iBot") with Cre-dependent expression of botulinum neurotoxin B light chain and GFP to mark recombined cells[28] (Fig. 1a). Botulinum neurotoxin B specifically cleaves and inactivates VAMP1, 2, and 3 (VAMP1/2/3) to prevent vesicle fusion with the plasma membrane (i.e., exocytosis) and has no other known targets[29,30] (Fig. 1b). We crossed iBot and *Cnp*-Cre mice to conditionally inactivate VAMP2/3 in pre-myelinating cells of the CNS and PNS[31], hereafter referred to as iBot;*Cnp*-Cre (Fig. 1a). Strikingly, iBot;*Cnp*-Cre mice were significantly smaller than their littermate controls (iBot/+) during the second postnatal week and rarely survived into the third postnatal week (Supplementary Fig. 1g–i). Given this early death, we first focused on an early-myelinating

region of the mouse CNS for our studies—the dorsal thoracic spinal cord.

We first quantified the specificity and penetrance of *Cnp*-Cre-driven iBot expression, as marked by GFP, using immunohistochemistry of spinal cord cross sections harvested at P12 (Supplementary Fig. 2a–c). On average, ~88% of iBot-expressing cells (GFP+) in iBot;*Cnp*-Cre mice were oligodendrocyte-lineage cells (Olig2+; GFP+); of these, ~74% were differentiated oligodendrocytes (CC1+; GFP+) while the remaining ~14% were oligodendrocyte precursors (Olig2+;GFP+;CC1−) (consistent with the known premature expression of *Cnp* promoter in a subset of oligodendrocyte precursors) (Supplementary Fig. 2d). Only 2% of iBot-expressing cells corresponded to neurons (NeuN+; GFP+) (Supplementary Fig. 2c, d). As a measure of penetrance, 79% of differentiated oligodendrocytes (CC1+) detectably expressed iBot (Supplementary Fig. 2e). The number of oligodendrocytes (differentiated and precursors) were not significantly affected (Supplementary Fig. 2f, g). Thus, iBot expression was highly penetrant and specific for oligodendrocytes without causing gross changes to their proliferation or survival.

To assess the role of VAMP2/3-mediated exocytosis in myelination, we first immunostained for myelin basic protein (MBP), a major structural component of myelin. Relative to littermate controls, iBot;*Cnp*-Cre exhibited a significant reduction in white matter area in the spinal cord (Fig. 1c, d and Supplementary Fig. 2h), brain cortex, and cerebellum (Supplementary Fig. 3), with no obvious effect on axonal abundance (Fig. 1e and Supplementary Fig. 2j). We next determined how myelin ultrastructure was affected in iBot;*Cnp*-Cre mice, using electron microscopy of the dorsal white matter of the thoracic spinal cord, which is comprised of parallel axon tracts[32] (Fig. 1f, g). iBot;*Cnp*-Cre mice were severely hypomyelinated, showing only 3% of axons with wrapped myelin compared to 28% of axons in the control at P12 (Fig. 1h). Furthermore, the wrapped axons in iBot;*Cnp*-Cre exhibited thinner myelin (higher *g*-ratio) than controls (Supplementary Fig. 2k, l). Interestingly, the iBot;*Cnp*-Cre and control samples showed no significant difference in the number of partially and fully ensheathed axons (defined as oligodendrocyte contact with at least half the circumference of an axon), suggesting that *Cnp*-Cre-driven inactivation of VAMP2/3 inhibits myelin wrapping but has less of an effect on the earlier process of axonal ensheathment (Fig. 1h). Surprisingly, when we examined myelination in the PNS, we found no gross changes in MBP staining or myelin wrapping by electron microscopy between sciatic nerve samples, despite prevalent iBot (GFP) expression in iBot;*Cnp*-Cre mice (Supplementary Fig. 4; see Discussion). All together, these data indicated that VAMP2/3-mediated exocytosis plays an essential role in CNS myelination.

### VAMP2/3-mediated exocytosis occurs preferentially in myelin sheaths
Since myelin wrapping and longitudinal growth are likely to require spatially regulated membrane addition, we asked if VAMP2/3-mediated exocytosis is spatially organized in oligodendrocytes. We transfected primary rat oligodendrocyte precursors with VAMP2- or VAMP3-pHluorin, a pH-sensitive variant of GFP that is quenched inside the acidic environment of a vesicle and fluoresces upon exocytosis[25,33]. In the absence of neurons, oligodendrocyte precursors differentiate in culture by extending numerous processes in the pre-myelinating stage and expanding into large, compact myelin membrane sheets as they mature[34,35] (Fig. 2a). We measured the frequency of exocytotic events per cell per minute, detected as a punctate increase and subsequent decay in fluorescence[25] (Fig. 2b and Supplementary Video 1). In differentiating oligodendrocytes, both VAMP2- and VAMP3-mediated exocytotic events occurred more frequently in the processes or myelin membrane sheets, as opposed to the soma (Fig. 2c and Supplementary Fig. 5a)[34,35]. Deacidification with $NH_4Cl$ revealed an abundance of VAMP2/3-vesicles in the soma that had not yet undergone exocytosis,

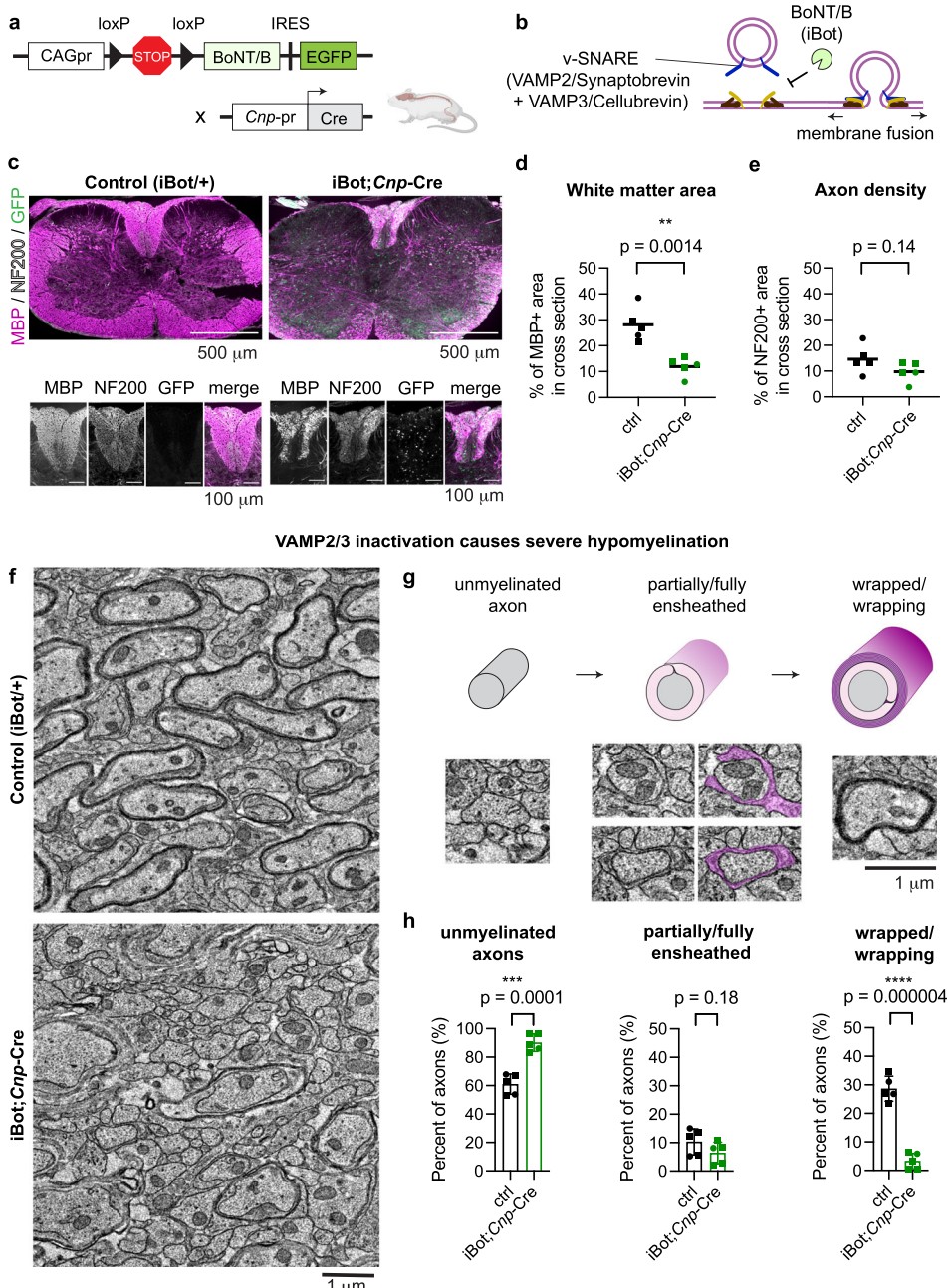

**Fig. 1 | VAMP2/3-mediated exocytosis is required for CNS myelination. a** Genetic cross for Cre-induced botulinum toxin light chain B (BoNT/B; or "iBot") under the constitutive CAG promoter (CAGpr) in pre-myelinating oligodendrocytes. Created with BioRender.com. **b** iBot inactivates v-SNAREs VAMP1 (not depicted), VAMP2, and VAMP3, blocking vesicular fusion to the plasma membrane (exocytosis). **c** Immunolabeling of P12 spinal cord cross sections from control (iBot/+, left) and iBot;*Cnp*-Cre (right) mice for MBP (magenta), neurofilament heavy chain 200 (NF200, white), and GFP to mark iBot expression (green). Scale bar, 500 μm. Bottom: insets of dorsal column containing parallel tracts of axons. Scale bar, 100 μm. See Supplementary Fig. 2h for MBP staining of *n* = 5 biologically independent replicates. **d** Quantification of white matter area defined as the percent of P12 spinal cord cross section area immunolabeled by the myelin marker MBP. Squares and circles denote males and females, respectively. Average ± SEM for *n* = 5; control: 28.1 ± 2.97%; iBot;*Cnp*-Cre: 11.9 ± 1.63%, statistical measurement (*p*-value) determined by an unpaired, two-tailed t-test. Source data are provided in the Source Data file. **e** Quantification of the percent of P12 spinal cord cross section area immunolabeled by the axon marker NF200. Squares and circles denote males and

females, respectively. Average ± SEM for *n* = 5; control: 14.6 ± 2.41%; iBot;*Cnp*-Cre: 9.76 ± 1.70%, *p*-value determined by an unpaired, two-tailed t-test. **f** Transmission electron microscopy images from P12 mouse spinal cord dorsal column cross sections (top: control (iBot/+); bottom: iBot;*Cnp*-Cre), with similar results for *n* = 5 biologically independent replicate quantified in **h**. Scale bar, 1 μm. Note the dramatic hypomyelination in iBot;*Cnp*-Cre mice. **g** Top: categories of myelination stages observed from cross sections of myelinated axons, with the axon in gray and myelin in magenta. Bottom: electron microscopy images of each myelination stage representative of *n* = 5 biologically independent replicates quantified in **h**. Scale bar 1 μm. **h** Quantification of the percent of axons in each myelination stage from electron microscopy in **f**, showing reduced wrapping and increased unmyelinated axons in iBot;*Cnp*-Cre mice. and circles denote males and females, respectively. Average ± SEM for *n* = 5; unmyelinated: control (61.0 ± 3.17%), iBot;*Cnp*-Cre (90.2 ± 2.76%); partially/fully ensheathed: control (10.3 ± 2.06%), iBot;*Cnp*-Cre (6.38 ± 1.68%); wrapped/wrapping: control (28.6 ± 1.96%), iBot;*Cnp*-Cre (3.28 ± 1.23%), *p*-value determined by an unpaired, two-tailed t-test. Source data are provided in the Source Data file.

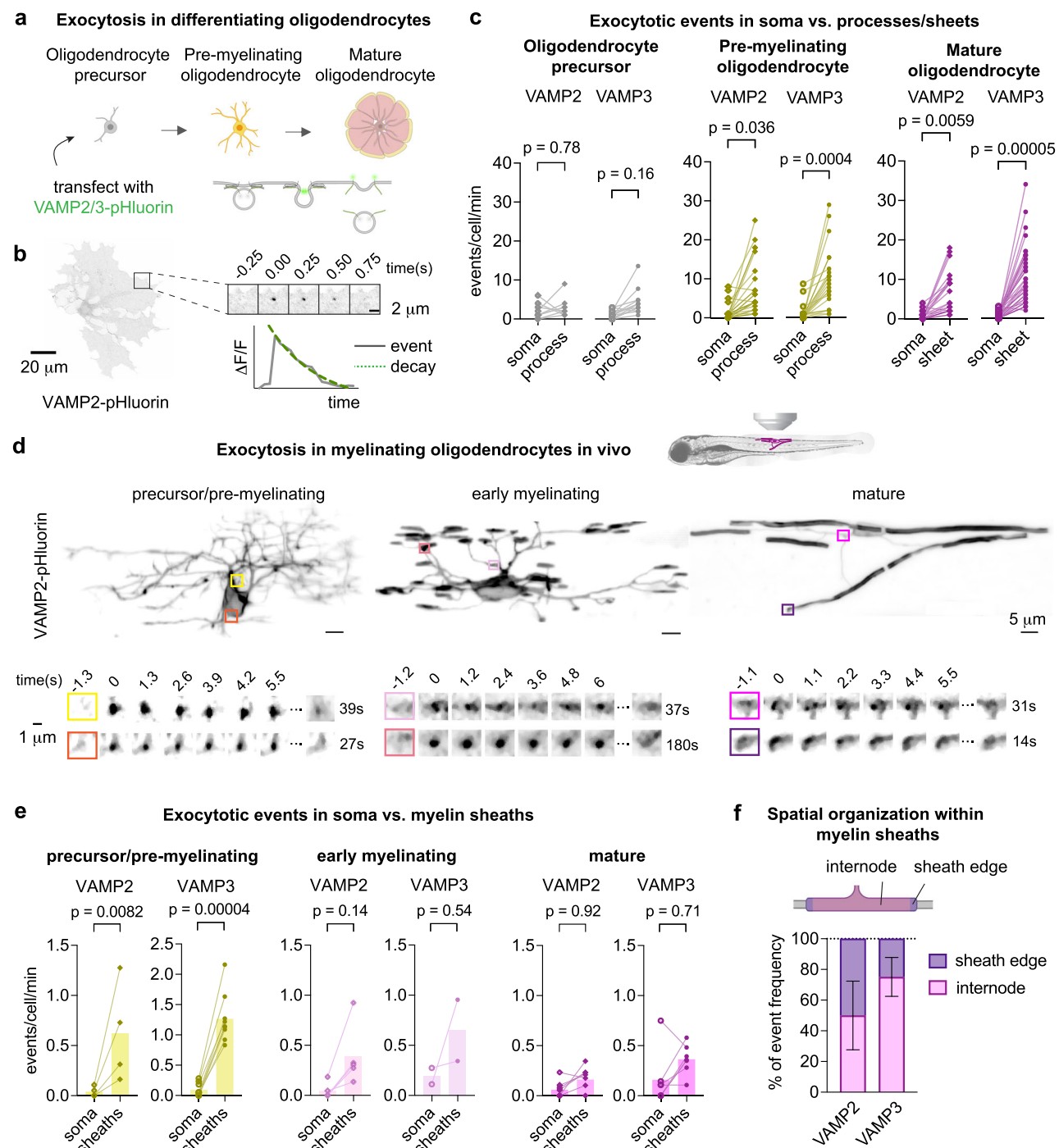

**Fig. 2 | VAMP2/3-mediated exocytosis occurs preferentially in myelin sheaths.**
**a** Top: diagram depicting oligodendrocyte differentiation in culture; bottom: transfection of oligodendrocyte precursors with pHluorin-tagged VAMP2 or VAMP3 to visualize exocytosis. Created with BioRender.com. **b** Representative image of a pre-myelinating oligodendrocyte expressing VAMP2-pHluorin. Montage shows an exocytotic event over time, and the corresponding plot of intensity vs. time shows the characteristic fluorescent increase and decay (fitted by the green dotted line). **c** Frequency of VAMP2- or VAMP3-pHluorin events localized to the soma or to the processes/sheets in cultured rat oligodendrocyte precursors (gray), pre-myelinating (yellow), or mature oligodendrocytes (magenta). Each pair of points connected by a line shows events for one cell. See Supplementary Table 1 for average event frequencies ± SEM. Statistical significance was determined using the mean of each biologically independent replicate (*n* = 3) by an ordinary one-way ANOVA with Tukey correction for multiple comparisons. Source data are provided

in the Source Data file. **d** Top: representative images of oligodendrocyte-lineage cells expressing VAMP2-pHluorin in the larval zebrafish spinal cord. Bottom: examples of VAMP2 exocytotic events within oligodendrocyte processes and sheaths. **e** Frequency of VAMP2- or VAMP3-pHluorin events in oligodendrocytes of the zebrafish spinal cord localized to the soma or to the processes/sheaths in precursors/pre-myelinating (yellow), early myelinating (pale pink), or mature oligodendrocytes (magenta). Each pair of points connected by a line shows events for one cell (typically from one fish). See Supplementary Table 2 for average event frequencies ± SEM. Statistical significance was determined by an ordinary one-way ANOVA with Tukey correction for multiple comparisons. **f** Spatial frequency of VAMP2 and VAMP3 exocytotic events within myelin sheaths, where the paranode is defined as 3 μm from the sheath edge[68]. The measured frequencies at paranodes (mean ± SEM) were (50 ± 22.4)% for VAMP2 and (24.9 ± 12.7)% for VAMP3, each with *n* = 25 sheaths from five experiments. See Supplementary Table 3.

suggesting that vesicles marked by VAMP2 and VAMP3 traffic away from the soma prior to exocytosis out in myelin sheets (Supplementary Fig. 5b). Consistent with prior work showing that overexpressed VAMP2-pHluorin in neurons localizes correctly and does not perturb exocytic function[36], neither the frequency of exocytic events nor oligodendrocyte cell area was affected by VAMP-pHluorin expression levels (Supplementary Fig. 5c, d).

To examine the spatial pattern of VAMP2/3 activity in myelin sheaths in vivo, we used live imaging of developing zebrafish expressing oligodendrocyte-targeted VAMP2- and VAMP3-pHluorin (Fig. 2d and Supplementary Video 2). The frequency of exocytosis was highest in the processes of oligodendrocyte precursors and pre-myelinating oligodendrocytes, consistent with the enrichment found in highly arborized pre-myelinating oligodendrocytes in culture (Fig. 2e, c). Of the exocytotic events within myelin sheaths, 50% of VAMP2 events occurred at the sheath edges (i.e., paranodes) (Fig. 2f), which was higher than the predicted frequency (34%) for uniform distribution of exocytosis (Supplementary Table 3). In contrast, VAMP3 events were uniformly distributed between the internode and sheath edges (Fig. 2f). Resolving how VAMP2- and VAMP3-mediated exocytosis are spatially regulated in sheaths remains an interesting question for future studies.

To determine if VAMP2/3-exocytosis contributes to membrane expansion, we determined the proportion of events that resulted in full-vesicle fusion rather than kiss-and-run exocytosis in cultured oligodendrocytes, where the outspread membrane allows for quantification of fluorescence spreading of exocytotic puncta. Full-vesicle fusion events integrate lipids and transmembrane proteins into the plasma membrane, and exhibit radial fluorescence spreading as pHluorin molecules diffuse from the initial site of fusion (Supplementary Fig. 5e, top). In contrast, kiss-and-run events recycle vesicles after secreting molecules with no net addition of membrane, and do not exhibit fluorescence spreading (Supplementary Fig. 5e, bottom)[25,26,37]. We distinguished full-vesicle fusion as events with radial fluorescence spreading, where the half-life of the bordering membrane fluorescence closely matched the half-life of the event center (Supplementary Fig. 5e–g). In pre-myelinating oligodendrocytes, full-vesicle fusion was the predominant mode of exocytosis for both VAMP2 and VAMP3 events, indicating an increased rate of membrane addition during early differentiation (Supplementary Fig. 5g, yellow). Based on our calculations (see Discussion), the frequency of full-vesicle fusion events we observed could, in theory, provide sufficient membrane for oligodendrocyte membrane growth during myelination. Thus, we next aimed to test the role of VAMP2/3 in myelin membrane expansion.

## VAMP2/3-mediated membrane expansion is required for myelin sheath elongation

Our discovery that VAMP2/3 mediate full-vesicle fusion in pre-myelinating oligodendrocytes suggested that VAMP2/3 may function to drive membrane expansion during myelination. To test this, we first purified primary oligodendrocyte precursors from neonatal brains of iBot;Cnp-Cre versus control littermates and induced differentiation in culture[38] (Fig. 3a). By the pre-myelinating stage (Day 3), 65% of iBot;Cnp-Cre cells expressed GFP as a marker of iBot expression, which increased to 80% of mature oligodendrocytes (Day 7) (Supplementary Fig. 6a, b). iBot;Cnp-Cre and control oligodendrocytes both produced arborized branches at the pre-myelinating stage (Day 3) with no difference in membrane area (Fig. 3b, c). Between Day 3 and Day 5 as oligodendrocytes formed myelin sheaths, control oligodendrocytes doubled their membrane area, consistent with the increase in full-vesicle fusion events we measured at this time point (Supplementary Fig. 5g). However, iBot;Cnp-Cre oligodendrocytes stalled in an

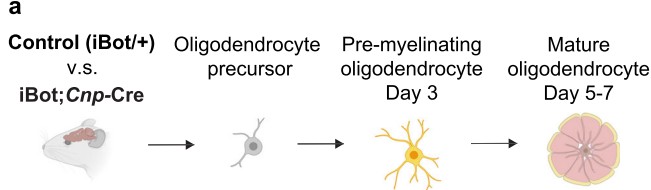

**a**

Control (iBot/+)  v.s.  iBot;*Cnp*-Cre

Oligodendrocyte precursor → Pre-myelinating oligodendrocyte Day 3 → Mature oligodendrocyte Day 5-7

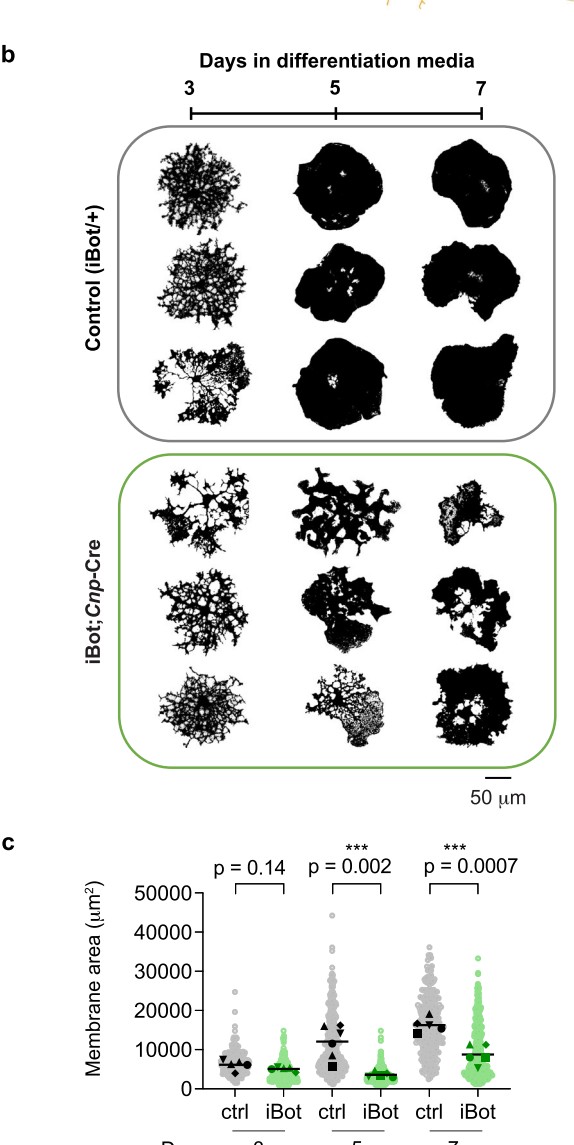

**b**   Days in differentiation media

3 5 7

Control (iBot/+)

iBot;*Cnp*-Cre

50 μm

**c**

p = 0.14 p = 0.002 *** p = 0.0007 ***

Membrane area (μm²): 50000, 40000, 30000, 20000, 10000

ctrl iBot ctrl iBot ctrl iBot
Day: 3 5 7

**Fig. 3 | VAMP2/3-mediated exocytosis is required for oligodendrocyte membrane expansion. a** Oligodendrocyte precursors purified from transgenic mouse brains (control vs. iBot;*Cnp*-Cre) and differentiated in culture to investigate cell-intrinsic effects of VAMP2/3 inactivation. Created with BioRender.com. **b** Thresholded masks of primary oligodendrocytes stained for GalCer lipid during a differentiation time course of 3, 5, and 7 days from control (top) and iBot;*Cnp*-Cre (bottom) mice to show cell morphologies. Scale bar, 50 μm. See Supplementary Fig. 6e for examples of GalCer staining. Source data are provided in the Source Data file. **c** Quantification of membrane surface area marked by GalCer lipid, where each light-shaded point corresponds to a single cell, each dark-shaded point represents the mean area of cells from one biologically independent replicate, and the line represents the mean area from *n* = 5 biologically independent replicates for each condition. See Supplementary Table 4 for descriptive statistics. Statistical measurements (*p*-value) were determined by an unpaired, two-tailed t-test. Source data are provided in the Source Data file.

arborized state and did not form full myelin sheets (Fig. 3b, c). After 7 days of differentiation, iBot;*Cnp*-Cre oligodendrocytes reached only half the area of control oligodendrocytes (-8700 µm² vs. -16,200 µm²; Supplementary Fig. 6e). iBot expression did not affect oligodendrocyte differentiation (as marked by GalCer lipid or MBP) or oligodendrocyte survival (Supplementary Figs. 6c, d, 7). Thus, inactivating VAMP2 and VAMP3 in pre-myelinating oligodendrocytes reduces membrane expansion without significantly affecting differentiation or cell survival.

We next asked whether VAMP2/3-mediated membrane expansion is necessary for myelin formation on axons. To investigate the initial stages of sheath formation, we used an oligodendrocyte-neuron co-culture system in which we seeded oligodendrocyte precursors purified from iBot;*Cnp*-Cre or control littermates onto dense arrays of axons generated by aggregating retinal ganglion cells (Fig. 4a, b and Supplementary Fig. 8)[39,40]. iBot;*Cnp*-Cre oligodendrocytes did not have a statistically significant difference in the number of sheaths per cell body compared to controls (Fig. 4c) but formed significantly shorter sheaths (Fig. 4d, average length of 60.5 µm in iBot;*Cnp*-Cre vs. 90.9 µm in controls), indicating a defect in lateral sheath elongation.

Is VAMP2/3-mediated membrane expansion necessary for myelin sheath elongation in vivo? We measured myelin sheath lengths in the corpus callosum and deep cortical layers at P12 (Fig. 4e, f), where myelin is still relatively sparse in the second postnatal week[41]. Average myelin sheath lengths were decreased in iBot;*Cnp*-Cre mice (39.6 µm) compared to littermate controls (54.2 µm) (Fig. 4g, h), corroborating the sheath elongation defect we observed in myelinating co-cultures of iBot-expressing oligodendrocytes (Fig. 4d). We estimated that shorter myelin sheaths (Fig. 4g) combined with a trend towards slightly fewer sheaths per oligodendrocyte (Fig. 4c) is sufficient to fully explain the high degree of unmyelinated stretches of axons we observed in electron microscopy cross-sections (Fig. 1f–h) in iBot;*Cnp*-Cre mice (see "Quantification of myelin coverage" in Methods).

Additionally, we tested the requirement of VAMP2/3 using a method orthogonal to iBot through the exogenous expression of dominant negative VAMP proteins (dn-VAMP2/3), in which the cytosolic portion of the v-SNARE blocks binding and fusion of endogenous v-SNAREs to the plasma membrane (Supplementary Fig. 9a)[42,43]. Expressing dn-VAMP2 or dn-VAMP3 individually in cultured oligodendrocytes was sufficient to reduce membrane expansion (Supplementary Fig. 9b–d). To determine if dn-VAMP2 affected myelin sheath elongation in vivo, we used adeno-associated virus (AAV)-mediated transgene expression to sparsely express dn-VAMP2 with the membrane marker GFP-caax under the oligodendrocyte-specific *Mbp* promoter (Supplementary Fig. 9e, f)[44–46]. In the mouse spinal cord at P12, dn-VAMP2 expression reduced sheath lengths (84.0 µm) compared to GFP-caax expression alone (110 µm) (Supplementary Fig. 9g–i). Together, our results from two orthogonal methods indicate that blocking exocytosis in pre-myelinating oligodendrocytes does not prevent their initial ensheathment of axons but results in shorter sheaths that leads to pronounced hypomyelination.

## Vesicles accumulate in the inner tongue of myelin sheaths upon inactivation of VAMP2/3

From finding an essential role for VAMP2/3 in oligodendrocyte membrane expansion and myelin sheath elongation, we next asked if and where membrane vesicles accumulated in oligodendrocytes upon inactivation of VAMP2 and VAMP3. High-resolution imaging revealed that GalCer-positive vesicular structures accumulated within stunted processes and along the membrane periphery of iBot;*Cnp*-Cre oligodendrocytes both in monocultures (Supplementary Fig. 10a) and in co-cultures with neurons (Supplementary Fig. 10b). In vivo, super-resolution microscopy showed that the majority of MBP signal from control samples appeared as paired, parallel tracks separated by an average distance of 0.89 ± 0.13 µm, which is within reported diameter

ranges for myelinated fibers in the corpus callosum (Fig. 5a, c)[47]. In contrast, iBot;*Cnp*-Cre sheaths exhibited bulges along MBP-stained tracks, which averaged a diameter of 1.46 ± 0.08 µm (Fig. 5a, c) and appeared similar to the vesicular structures observed in oligodendrocyte-neuron co-cultures (Supplementary Fig. 10b). Super-resolution microscopy of iBot;*Cnp*-Cre spinal cord cross sections revealed multiple enclosed vesicular structures within a bulged myelin sheath (Fig. 5b and Supplementary Videos 3, 4). By electron microscopy, we found structures resembling enlarged vesicles in 31% of myelin cross sections from iBot;*Cnp*-Cre, as opposed to only 8% in controls (Fig. 5d, e). The accumulated vesicles in iBot;*Cnp*-Cre most frequently occurred at the inner tongue (innermost layer) of the myelin sheath, suggesting that VAMP2/3 normally mediate vesicle fusion at the axon-myelin interface (Fig. 5d, e). Interestingly, these large, pleomorphic vesicles, observed by both super-resolution fluorescence microscopy and electron microscopy, reached up to 0.5 µm in diameter, suggesting that, in the absence of VAMP2/3, vesicles may instead undergo homotypic fusion, which is known to form large secretory granules[48] in a VAMP2/3-independent manner[49] and has been postulated to occur during myelin wrapping[15] (See Discussion).

## Oligodendrocyte VAMP2/3 is required for delivery of myelin adhesion proteins and node of Ranvier formation

Given the spatial organization of VAMP2/3 activity within internodes, we next asked which myelin-associated proteins depend on VAMP2/3-mediated membrane trafficking. We covalently labeled surface proteins in cultured control and iBot;*Cnp*-Cre oligodendrocytes using NHS-biotin and immunoprecipitated surface-biotinylated proteins and membrane-proximal interactors for mass spectrometry (Fig. 6a and Supplementary Fig. 12a). In parallel, control unbiotinylated oligodendrocytes were analyzed to exclude non-specific interactions. We identified 123 proteins that were specifically depleted from iBot;*Cnp*-Cre oligodendrocytes (Fig. 6b and Supplementary Table 5). 98% of the VAMP2/3-dependent hits had also been detected in acutely isolated oligodendrocytes from mouse brains and from purified myelin, validating the ability of primary culture to recapitulate in vivo oligodendrocytes[24,50,51]. Gene ontology analysis revealed a significant enrichment of proteins that localize to myelin sheaths, paranodes, cell-cell junctions, and vesicles (Fig. 6c).

The paranode consists of specialized axon-myelin junctions that flank the nodes of Ranvier and are essential for action potential propagation[52,53]. Notably, top VAMP2/3-dependent hits included myelin adhesion proteins that normally localize to the oligodendrocyte-axon interface: contactin-1 (Cntn1), neurofascin (Nfasc), and myelin-associated glycoprotein (MAG) (Fig. 6d). Cntn1 and glial Nfasc (Nfasc-155) are required for the establishment of axon-myelin junctions at paranodes, and MAG is required to maintain axon-myelin interactions at internodes[54–58]. Surface staining of MAG was reduced in iBot;*Cnp*-Cre oligodendrocytes, validating MAG as a substrate of VAMP2/3-mediated exocytosis (Supplementary Fig. 12b–d). Other VAMP2/3-dependent hits included intracellular membrane-proximal proteins such as MBP and ankyrin-G (AnkG), a cytoskeletal scaffolding protein that directly interacts with Nfasc-155 in oligodendrocytes, where it is required for paranode assembly[59].

Finally, given that VAMP2/3 mediates myelin sheath elongation and the surface insertion of myelin adhesion proteins with known roles in regulating axoglial junctions, we asked if oligodendrocyte VAMP2/3 is required for node of Ranvier assembly. We immunostained longitudinal sections of the spinal cord for contactin-associated protein (Caspr), an axonal membrane protein enriched at paranodal regions, and AnkG, which (in addition to its aforementioned role at the paranode) anchors sodium channels at nodes of Ranvier (Fig. 6e)[53,60]. Mature nodes of Ranvier appear as puncta of AnkG flanked by Caspr on both sides (Fig. 6f). Immature nodes include heminodes, which consist of adjacent, singular clusters of Caspr and AnkG, or Caspr clusters

lacking flanked AnkG (Fig. 6f). Control spinal cord white matter showed a nearly equal distribution of mature nodes, heminodes, and other Caspr clusters at P12 (Fig. 6g), which is consistent with the nodal distribution reported for early CNS myelination in mice[52]. In iBot;*Cnp*-

Cre mice, we found a severe reduction in the formation of Caspr or AnkG clusters (Fig. 6g). Consistent with the premature death of iBot;*Cnp*-Cre mice observed two weeks after birth (Supplementary Fig. 1i), mice lacking node of Ranvier formation die between P12 and

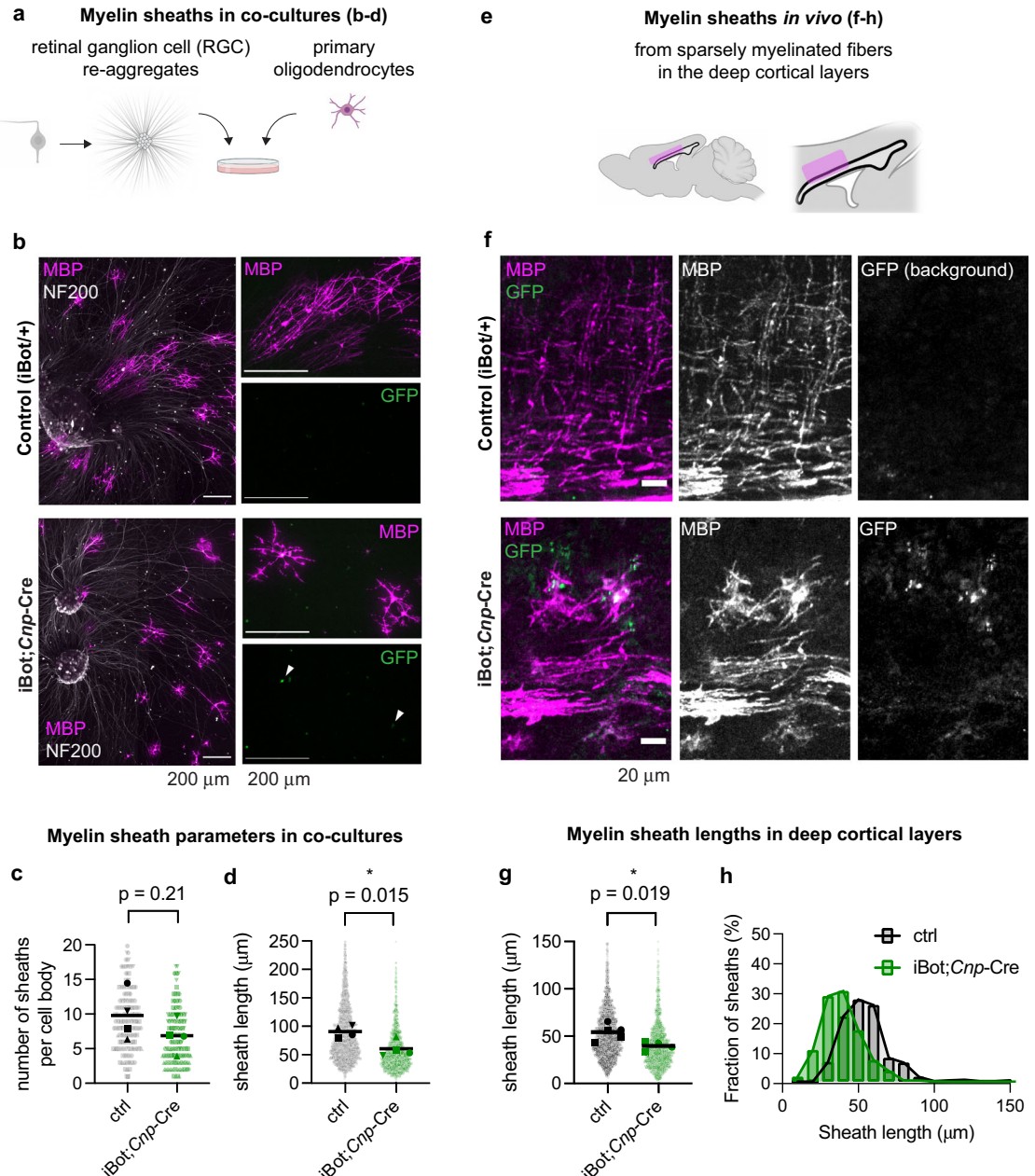

**Fig. 4 | VAMP2/3-mediated membrane expansion is required for myelin sheath elongation. a** Schematic of co-cultures between oligodendrocytes (magenta) and CNS-derived axons from retinal ganglion cell aggregates with radially protruding axons (gray). Created with BioRender.com. **b** Primary oligodendrocytes purified from control (top) or iBot;*Cnp*-Cre mouse brains (bottom) cultured on CNS-derived axons (retinal ganglion cell re-aggregates) for 7 days and stained for MBP (magenta) and NF200 (white), with GFP to mark iBot expression (green). All scale bars, 200 μm. See Supplementary Fig. 8 for examples of sheath quantification. Source data are provided in the Source Data file. Quantification of the number of sheaths per oligodendrocyte (**c**) and the length of sheaths (**d**) from co-cultures, where each light-shaded point represents a single sheath, each dark-shaded point represents the mean length of all sheaths from one biologically independent replicate, and the line represents the mean sheath length from $n = 4$ biologically independent replicates with 33–63 cells each. Mean number of sheaths per cell body ± SEM: control 9.8 ± 1.8, iBot 6.9 ± 1.2. Mean sheath length ± SEM: control 90.9 ± 5.2 μm, iBot

60.5 ± 7.4 μm. Statistical measurements (*p*-value) were determined by an unpaired, two-tailed t-test. Source data are provided in the Source Data file. **e** Schematic of a brain region containing sparsely myelinated fibers at P12 examined in **f**–**h**. Created with BioRender.com. **f** Immunolabeling of P12 mouse sagittal brain slices from control (top) and iBot;*Cnp*-Cre (bottom) littermates for MBP (magenta) and GFP (green). Control samples do not show GFP signal, since iBot is not expressed. Images are representative of $n = 5$ control and $n = 4$ iBot;*Cnp*-Cre mice quantified in **g**, **h**. Scale bar, 20 μm. Source data are provided in the Source Data file. Quantification of sheath length from P12 brain slices depicting scatter plots (**g**) and histogram distribution (**h**) between control (gray) and iBot;*Cnp*-Cre (green). Squares and circles denote males and females, respectively. Mean sheath length ± SEM: control 54.2 ± 3.81 μm, iBot 39.6 ± 2.38 μm from $n = 5$ (control) or 4 (iBot;*Cnp*-Cre) biologically independent replicates with 441–472 total sheaths quantified per condition. Statistical measurement (*p*-value) was determined by an unpaired, two-tailed t-test. Source data are provided in the Source Data file.

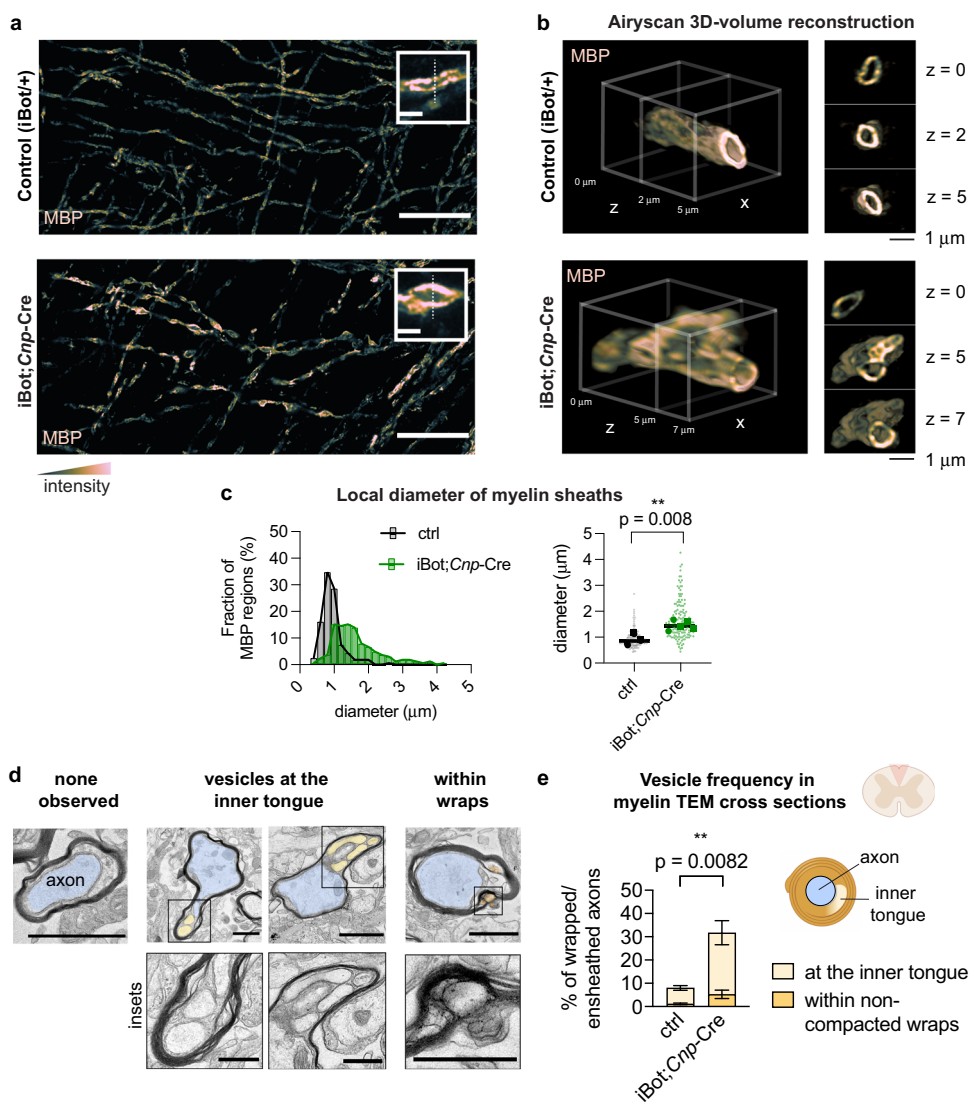

**Fig. 5 | Vesicles accumulate at the inner tongue of myelin sheaths following inactivation of VAMP2/3. a** Super-resolution (Airyscan confocal) images of P12 mouse cingulate cortex from control (top) and iBot;*Cnp*-Cre (bottom) littermates immunostained for MBP. Scale bar, 20 μm (inset, 2 μm). **b** 3D reconstructed Airyscan confocal z-stack of a myelinated axon cross section from a control (top) and iBot;*Cnp*-Cre (bottom) mouse spinal cord immunostained for MBP. See also Supplementary Videos 3 and 4. **c** Quantification of sheath diameter on thresholded regions of high MBP intensity from **a**. Squares and circles denote males and females, respectively. Mean sheath diameter ± SEM: control 0.89 ± 0.13 μm from 161 regions quantified within *n* = 3 biologically independent replicates; iBot 1.46 ± 0.08 μm from 165 regions quantified within *n* = 5 biologically independent replicates. See Supplementary Fig. 11 for quantification methodology. Statistical

measurement (p-value) was determined by an unpaired, two-tailed t-test. Source data are provided in the Source Data file. **d** Top: electron microscopy of myelinated axons from control and iBot;*Cnp*-Cre spinal cords, where the axon is colored blue and myelin vesicles are colored in yellow/orange (scale bar 1 μm). Bottom: insets for myelin vesicles from top row (scale bar 0.5 μm). Images are representative of myelinated axons observed from *n* = 5 biologically independent replicate pairs. **e** Quantification of electron microscopy data from mouse spinal cord cross sections for the presence of vesicles within myelinated axons. Control vs. iBot;*Cnp*-Cre mean ± SEM, respectively from *n* = 5 biologically independent replicates: at the inner tongue 6.87 ± 1.01% vs. 26.5 ± 5.19%; within non-compacted wraps 1.12 ± 0.34% vs. 5.24 ± 1.80%, *p*-values determined by an unpaired, two-tailed t-test. Created with BioRender.com. Source data are provided in the Source Data file.

P18[52,54,61]. Thus, VAMP2/3-mediated exocytosis acts as an indispensable mechanism by which oligodendrocytes regulate node of Ranvier assembly. Together, our results indicate that VAMP2/3-mediated exocytosis is required in oligodendrocytes for myelin wrapping, sheath elongation, and formation of nodes of Ranvier (Fig. 7).

## Discussion

How new membrane is added to power myelination is an important, unanswered question in neurobiology. In this study, we identify VAMP2/3-mediated exocytosis as a critical mechanism for membrane expansion during myelin sheath formation. By integrating cellular and in vivo approaches, we discovered spatial organization of VAMP2/3-mediated exocytosis in myelin sheaths that delivers both membrane

material and select adhesion proteins to the paranodes and inner tongue. Our mass spectrometry data identified myelin proteins that were delivered by oligodendrocyte-regulated VAMP2/3 exocytosis, positioning key myelin-axon adhesion proteins, intracellular junction scaffolding proteins, and vesicle sorting components to establish oligodendrocyte-axon interactions. Inactivating VAMP2/3 in pre-myelinating oligodendrocytes prevented them from initiating myelin wrapping and inhibited their ability to elongate along axons, leading to thinner, shorter sheaths (Supplementary Fig. 13a). Myelin sheath thickness and length are increasingly recognized to be dynamic regulatory parameters for information processing in the CNS[7,10,62,63], but little is known about the cellular mechanisms controlling activity-dependent membrane remodeling in oligodendrocytes. Together, our

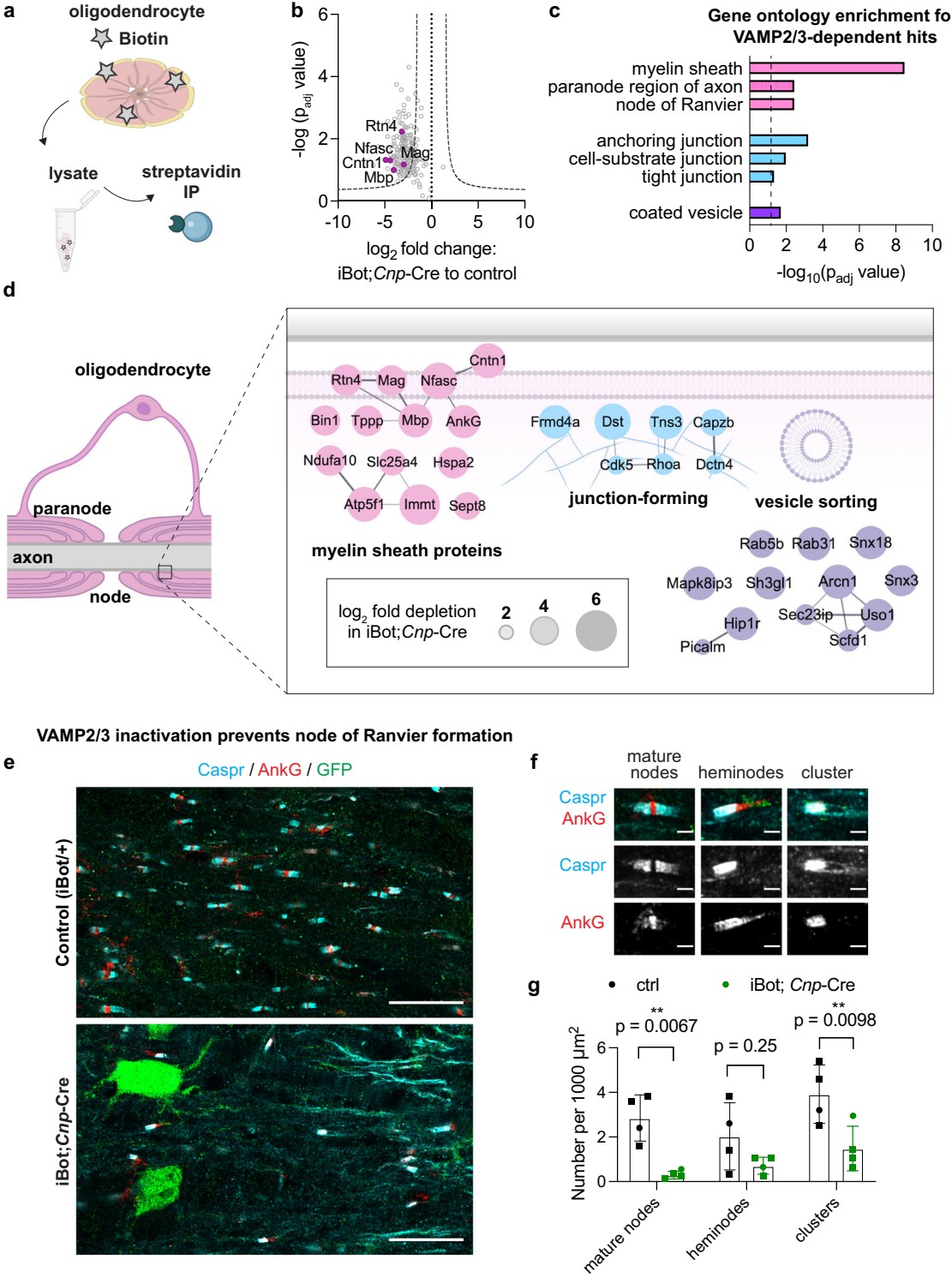

**VAMP2/3 inactivation prevents node of Ranvier formation**

findings suggest that developmental and experience-dependent regulation of myelination may converge on coordinating VAMP2/3-mediated exocytosis in oligodendrocytes.

Our results raise several questions. First, how much membrane does VAMP2/3-mediated exocytosis contribute, and is it likely that this is the sole mechanism of oligodendrocyte membrane expansion during myelination? In primary oligodendrocyte cultures—where the entire membrane is visible in 2D—we measured that a pre-myelinating oligodendrocyte adds an average of 6000 $\mu m^2$ over 48 h, similar to in vivo estimates[4]. During this phase of rapid membrane expansion, cultured oligodendrocytes exhibit an average of 23 exocytotic events

per minute for VAMP2 and VAMP3 combined, with 80% resulting in full-vesicle fusion. Assuming spherical vesicles with a diameter range of 100–200 nm[19,25], VAMP2/3-mediated exocytosis would add 1670–6650 $\mu m^2$ over 48 h, accounting for 27–110% of the added surface area. Thus, VAMP2/3-mediated exocytosis could theoretically provide sufficient new membrane for myelination, at least during the early stages of myelination that are modeled in culture. However, neurons can induce trafficking of myelin proteolipid protein (PLP1) through an orthogonal exocytic pathway in culture, likely via VAMP7[22,64]. The absence of PLP1 from our mass spectrometry results (Supplementary Table 5) is consistent with a VAMP2/3-independent

**Fig. 6 | Oligodendrocyte VAMP2/3 is required for delivery of myelin adhesion proteins and node of Ranvier formation. a** Biotinylation of surface proteins on mature oligodendrocytes differentiated in culture followed by immunoprecipitation (IP) with streptavidin beads for mass spectrometry. Created with BioRender.com. **b** Proteomic analysis of immunoprecipitated surface proteins and interactors from control vs. iBot;*Cnp*-Cre oligodendrocytes. Data points exclude non-specific interactors of the streptavidin IP identified in the non-biotinylated samples. Statistical measurements were determined from two-sample T-test with a Benjamini-Hochberg false discovery rate adjustment ($p_{adj}$). The negative $\log_{10} p_{adj}$ of each specific interactor was plotted against its average $\log_2$ fold change between iBot;*Cnp*-Cre and control samples. The dotted curve corresponds to the threshold cutoff for a 2-fold change between samples and a permutation-based false discovery rate correction for multiple comparisons with $p < 0.05$. Magenta points highlight notable myelin proteins that are significantly dependent on VAMP2/3 for surface delivery. **c** Selected gene ontology (GO) annotations using gProfiler for VAMP2/3-dependent hits from the surface proteomic analysis (**b**) with a dotted line marking $p_{adj} = 0.05$. Adjusted *p*-values were determined by the gProfiler website,

which uses the cumulative hypergeometric test with the g:SCS correction method. **d** Annotated localization of top VAMP2/3-dependent proteins in oligodendrocytes. The area of each circle scales linearly to the $\log_2$ fold depletion in iBot;*Cnp*-Cre oligodendrocytes. Lines connecting proteins denote reported and predicted protein interactions from the STRING database. Created with BioRender.com. **e** Immunohistochemistry of axonal node components Caspr (cyan) and AnkG (red) for longitudinal sections of the spinal cord harvested from control (top) and iBot;*Cnp*-Cre (bottom) littermates at P12. Images are representative of *n* = 4 biologically independent replicates quantified in **g**. Scale bar, 20 μm. **f** Representative images of node classifications. Scale bar, 2 μm. **g** Quantification of the number of nodes in each category over 48,000–72,000 μm$^2$ of spinal cord longitudinal section for each of *n* = 4 biologically independent replicates. Square and circle symbols denote males and females, respectively. Mean number of nodes per 1000 μm$^2$ ± SEM for control vs. iBot;*Cnp*-Cre, respectively: mature nodes 2.85 ± 0.52 vs. 0.28 ± 0.09; heminodes 2.03 ± 0.76 vs. 0.70 ± 0.19; clusters 3.93 ± 0.66 vs. 1.48 ± 0.50. Statistical significance was determined by one-way ANOVA with multiple comparisons correction. Source data are provided in the Source Data file.

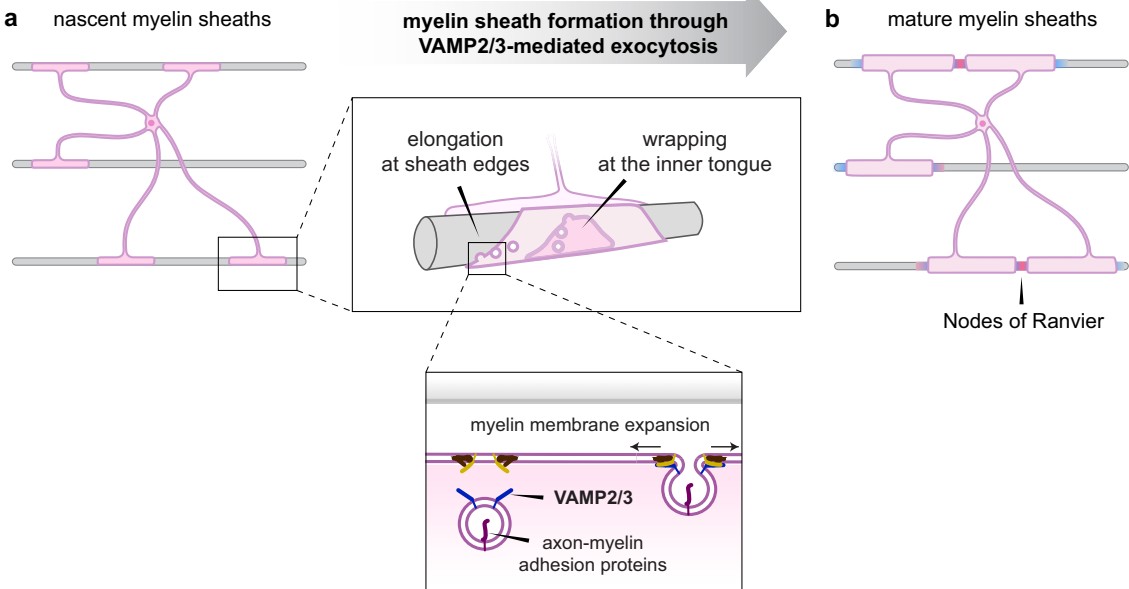

**Fig. 7 | Model figure of how VAMP2/3-mediated exocytosis drives CNS myelination. a** In nascent myelin sheaths, VAMP2/3-mediated exocytosis occurs at the inner tongue to drive wrapping and at sheath edges to drive elongation. VAMP2/3-mediated exocytosis results in full-vesicle fusion that incorporates myelin

membrane material and axon-myelin adhesion proteins. Created with BioRender.com. **b** VAMP2/3-mediated exocytosis coordinates sheath elongation with surface delivery of axon-myelin adhesion proteins to form nodes of Ranvier in the CNS. Created with BioRender.com.

mechanism of trafficking for PLP1. Such VAMP2/3-independent transport mechanisms may be necessary to spatially partition membrane proteins for axon-myelin adhesion at paranodes from myelin-myelin adhesion between wraps and may contribute additional membrane for myelination.

How might neuronal activity regulate membrane dynamics in myelin? One exciting possibility is that neuronal activity could stimulate VAMP2/3-mediated exocytosis in oligodendrocytes. Both VAMP2 and VAMP3 can drive exocytosis in a Ca$^{2+}$-dependent manner[27]. Recent studies identified local Ca$^{2+}$ transients in nascent myelin sheaths in response to neuronal activity or axonal release of glutamate, and provided evidence that these Ca$^{2+}$ transients influence myelin sheath length[65–67]. Moreover, glutamate release from axons selectively occurs adjacent to growing myelin sheaths and consequently stimulates elongation[68]. By live imaging exocytosis in vivo, we found that oligodendrocyte VAMP2 events were preferentially distributed at sheath edges (Fig. 2f), consistent with a model in which local axonal release of glutamate could spatially restrict oligodendrocyte exocytosis to drive sheath elongation. Moreover, electron microscopy revealed that

VAMP2/3-mediated exocytosis along internodes occurs at the innermost layer, positioning VAMP2/3 at the axon-myelin interface. VAMP2/3-mediated exocytosis in oligodendrocytes may thus be the downstream driver that couples activity-induced Ca$^{2+}$ transients to myelin membrane addition. However, the extent to which Ca$^{2+}$ transients in oligodendrocytes depend on neuronal activity is debated[69], so this remains an important question for future work.

Long-term stability of axo-glial units would also depend on maintaining myelin adhesion at paranodes and between wraps. Does VAMP2/3-mediated exocytosis contribute to myelin maintenance? Recent studies report the breakdown of myelin sheaths upon inducing the genetic ablation of PLP1 or MBP during adulthood, suggesting that continuous transport of new material is necessary for myelin maintenance[70,71]. Single-cell transcriptomics of adult and aged mice revealed that VAMP2 and VAMP3 continue to be the highest expressed v-SNAREs in oligodendrocytes through the lifetime of a mouse[72] (Supplementary Fig. 13b). Our current study is limited by the premature death of iBot;*Cnp*-Cre at 2–3 weeks, but iBot expression in adult oligodendrocytes may be inducible through *Pdgfrα*-CreERT or

*Plp1*-CreERT. Thus, whether VAMP2 and VAMP3 also play an essential role in myelin maintenance is an interesting question for future investigation.

Furthermore, what mechanisms power membrane expansion for PNS myelination? Surprisingly, we found no evidence that PNS myelination requires VAMP1/2/3, despite strong expression of iBot in Schwann cells (Supplementary Fig. 4). Earlier studies had also suggested that Schwann cells and oligodendrocytes may use distinct mechanisms for membrane addition during myelination. Pulse labeling of lipids and glycoproteins in PNS myelin demonstrated that new myelin material appeared first at the outer wraps and then later in the inner wraps[73,74]. In contrast, pulse-labeled glycoproteins in CNS myelin appeared first at the inner wraps and later at the outer wraps[5]. Thus, dedicated vesicle trafficking to the inner tongue may not be required for myelination in Schwann cells. Alternatively, vesicle trafficking in Schwann cells may rely on v-SNAREs insensitive to iBot cleavage. Recent RNA sequencing data from sciatic nerves revealed that the expression levels of iBot-insensitive *Vamp4, Vamp7*, and *Vamp8* are comparable to *Vamp2* in myelinating Schwann cells[75]. Understanding how membrane expansion differs between PNS and CNS myelination may offer insight into why myelin can robustly regenerate in the PNS but not in the CNS.

Finally, what is the role of SNARE-mediated exocytosis in remyelination? As the loss of myelin causes cumulative nerve damage in multiple sclerosis (MS), repair of myelin is hindered by pre-myelinating oligodendrocytes that fail to wrap or only generate thinner, shorter sheaths[76-78]. Ribosome sequencing of a cuprizone-induced mouse model for MS showed that VAMP3 was the only v-SNARE with increased translation in oligodendrocytes during the initial remyelination phase[79] (Supplementary Fig. 13c). Moreover, 34 of the VAMP2/3-dependent substrates and interactors we identified were also upregulated during initial remyelination, including MAG, Nfasc, and Rab31 (Supplementary Table 6), suggesting active VAMP3-mediated exocytosis during acute remyelination. However, in human MS patient samples, *VAMP3* mRNA is downregulated in differentiation-committed oligodendrocytes (COP) across all MS lesion types relative to normal-appearing white matter, suggesting deficient VAMP3 levels in MS pathology (Supplementary Fig. 13d)[80]. Our findings suggest the possibility that restoring VAMP2/3-mediated exocytosis in oligodendrocytes may represent a therapeutic avenue to promote remyelination.

In summary, our findings demonstrate that VAMP2/3-mediated exocytosis drives membrane addition in oligodendrocytes to power myelination, raising several questions about its role and regulation in myelin plasticity, maintenance, and regeneration. Future studies of VAMP2/3-mediated exocytosis in myelin will further our understanding of the diverse modes and functions of regulated exocytosis in the nervous system and potentially identify specific modulators of membrane expansion that act as therapeutic targets to restore white matter function in neurological disorders.

### Limitations of this study and alternative interpretations

One limitation of our study is that the *Cnp*-Cre transgenic mouse line has been shown to mediate recombination in some classes of neurons[81,82]. Since suppressing exocytosis in neurons (e.g., synaptic vesicle release) has been shown to inhibit myelination in a cell-non-autonomous manner[68,83,84], this raises the question of whether part of the myelination phenotypes we observed could be due to "leaky" expression of Cre in neurons. However, we found that only ~2% of iBot-expressing cells were neurons (NeuN+; GFP+), which is unlikely to account for the magnitude of myelin defects we observed (Supplementary Fig. 2d). In support of this interpretation, iBot expression in oligodendrocyte-lineage cells using two different Cre lines (NG2-Cre and PDGFRα-CreERT) caused similar myelin defects as iBot;*Cnp*-Cre, including decreased white matter area and hypomyelination[85,86].

Although we cannot exclude that iBot expression outside of oligodendrocyte-lineage cells contributed to reduced mouse body weight and premature death, mice mutants that are unable to assemble nodes of Ranvier in the CNS exhibited premature death within 2–3 weeks of birth, consistent with the phenotype and consequence of our iBot;*Cnp*-Cre mice[52]. Together with our orthogonal AAV-mediated expression of dominant negative v-SNAREs (Supplementary Fig. 9), our results argue that oligodendrocyte VAMP2/3-dependent exocytosis is required for myelination in the rodent CNS.

Our data and interpretation support a model in which VAMP2/3 directly drive membrane addition but do not exclude indirect mechanisms that contribute to myelin growth. For instance, we hypothesized that large vesicular structures form in iBot;*Cnp*-Cre myelin due to blocked exocytosis (Fig. 5). An alternative explanation may be that these vesicular structures occur due to loss of myelin membrane integrity caused by the depletion of membrane proteins, as observed in oligodendrocytes lacking connexins and potassium channels[87-89]. Future studies to obtain a time course of vesicle appearance in vivo may distinguish between unfused vesicles in transit and vacuolization. In addition, cultured iBot-expressing oligodendrocytes eventually expand in membrane area (albeit at a slower rate than wild-type oligodendrocytes). Potential explanations include the possibility that the cell-intrinsic upregulation of VAMP2/3 during differentiation outcompetes iBot activity, leading to incomplete cleavage of VAMP2/3 at later stages, or that VAMP2/3-independent mechanisms of membrane addition (such as the iBot-insensitive VAMP7 discussed above) may compensate at later stages. Although our study has uncovered critical roles for VAMP2/3 in membrane addition and adhesion protein localization, VAMP2/3-mediated exocytosis may also regulate myelination through the proper localization of cell-surface receptors necessary for growth factor signaling and/or the secretion of soluble signals[86]. How oligodendrocyte exocytosis mediates intercellular communication among glia and neurons, and how those mechanisms may be coupled to myelin membrane expansion, remains a ripe area for future investigation.

## Methods

### Mouse experiments

All rodent procedures were approved by Stanford University's Administrative Panel on Laboratory Animal Care (APLAC, Protocol 32260) and followed the National Institutes of Health guidelines. Mice were group-housed under standard 12:12 light-dark cycles at temperatures of 18–23 °C and 40–60% humidity with free access to food and water, and disposable bedding in plastic cages. All mice were regularly monitored by veterinary and animal care staff and were not involved in prior procedures or testing. Sprague-Dawley rats and C57BL/6 mice were ordered from Charles River Laboratories. Both male and female mice were studied for all in vivo experiments.

The transgenic iBot mice line, which contains the floxed-STOP site upstream of the *Clostridium botulinum* neurotoxin light chain followed by an IRES-GFP, was kindly gifted by Dr. Frank Pfrieger (University of Strasbourg) and Dr. Shane Hentges (Colorado State University). The previously created *Cnp1*-Cre mouse line[31] was a kind gift from Dr. Klaus Nave (Max Planck Institute for Experimental Medicine) and was maintained as heterozygotes by breeding with C57BL/6 mice. Males and females were pooled for analysis because no sexual dimorphism for myelination defects were observed after the conditional expression of iBot.

To establish new conditional mouse lines, heterozygous iBot/+ mice were crossed with heterozygous *Cnp1*-Cre/+ mice. Offspring were genotyped using the following primers: Cre allele (forward): 5′-GCTAAGTGCCTTCTCTACACCTGC-3′, (reverse): 5′-GGAAAATGCTTCTGTCCGTTTG-3′; eGFP (forward): 5′- CGTGTTCCACTCGAAGAGTT-3′, (reverse): 5′- GGCAAAACTTCATTTGCATT −3′. For further reference, *Cnp1-Cre/+;iBot/+* is referred to as iBot;*Cnp*-Cre.

For histological analysis, mice were anesthetized by injecting a cocktail of ketamine (100 mg/kg) and xylazine (20 mg/kg). The animals were then transcardially perfused with PBS and then with 4% paraformaldehyde (PFA) (diluted into 1× PBS from 16% PFA, Electron Microscopy Sciences), followed by post-fixation with 4% PFA for immunohistochemistry or Karlsson and Schultz (KS) buffer[90] for transmission electron microscopy.

## Antibodies for immunostaining

The following primary antibodies were used at the specified dilution: rat anti-MBP (Abcam ab7349, 1:100 for tissue, 1:400 for cultured cells; knockout-validated[35], rabbit anti-Olig2 (Sigma AB9610, 1:1000), mouse anti-CC1 (Oncogene OP80, 1:100), mouse anti-NeuN (Clone A60, EMD Millipore MAB377, 1:1000), rabbit anti-Caspr (Abcam ab34151, 1:1000), mouse anti-AnkG (Sigma MABN466, 1:500), mouse anti-NF200 (Sigma N0142, 1:100), chicken anti-GFP (Abcam 13970, 1:1000), mouse-anti MAG clone 513 (EMD Millipore, 1:20 for surface staining, 1:100 for total MAG), and mouse anti-galactosylceramide (GalCer) hybridoma[38,91], 1:50).

The following secondary antibodies were used at a 1:1000 dilution for tissue and cell staining: donkey anti-rat Alexa Fluor 594 (Thermo Scientific A-21209), goat anti-rat Alexa Fluor 647 (Thermo Scientific A-21247), donkey anti-rabbit Alexa Fluor 647 (Thermo Scientific A-31573), donkey anti-mouse Alexa Fluor 594 (Thermo Scientific A-21203), donkey anti-mouse Alexa Fluor 647 (Thermo Scientific A-31571), and goat anti-chicken Alexa Fluor 488 (Thermo Scientific A-32931).

## RNAscope for fluorescent in situ hybridization

The protocols for sample preparation, probe hybridization, and imaging were adapted from manufacturer guidelines for the RNAscope HiPlex8 Reagent Kit (Advanced Cell Diagnostics, #324100). Briefly, freshly dissected spinal cords from P10 mice were embedded and frozen in O.C.T. The frozen tissues were sliced into 20 μm thin cross sections using a cryostat (CM3050 S, Leica Microsystems), dried onto Superfrost Plus slides (VWR) at −20 °C, and stored at −80 °C without fixation.

Prior to probe hybridization, samples were fixed in 4% PFA for 60 min at RT and then dehydrated through washes of 50, 70, and 100% ethanol (v/v in water) for 5 min each. The final 100% ethanol wash was allowed to air dry for 5 min at RT. Samples were treated with Protease IV reagent for 30 min at RT and then washed twice with 2 min incubations in 1× PBS.

All target probes were ordered from Advanced Cell Diagnostics, unless otherwise indicated. Samples were incubated with pooled target probes against *Mus musculus Olig2, Pdgfra, Vamp2, Aspa*, and *Vamp3* mRNA (#447091-T1, #480661-T2, #573151-T3, #425891-T5, and #573161-T6, respectively) for 2 h at 40 °C. Unbound probes were washed off with 1× PBS in two 2 min incubations. For the first round of target probe visualization, probes were amplified with 30 min incubations at 40 °C in RNAscope HiPlex solutions Amp1–3 with two 2 min washes. After the final washes, selected secondary fluorescent probes (RNAscope Fluoro T1–T3) targeting *Olig2, Pdgfra* and *Vamp2* were incubated with the sample at 40 °C for 15 min. Excess probe was washed off with two 2 min washes and counterstained with DAPI for 30 s at RT. Samples were treated with ProLong Gold Antifade Mountant (Life Technologies #10144), covered with coverslips, and imaged by confocal microscopy (Zeiss LSM 880) with 20×/NA0.80 objective lens. Slides were stored in the dark at 4 °C overnight.

Prior to the second round of probe hybridization, the RNAscope Fluoro T1-T3 fluorophores were cleaved according to manufacturer's protocols. In summary, the slide was incubated in 4× SSC (RNAscope kit) at RT for 30 min until the coverslips detached without force. The samples were then washed again with 4× SSC and then treated with freshly diluted 10% cleaving solution in 4× SSC (RNAscope kit) for 15 min at RT, and then washed with two 2 min incubations in 0.5%

Tween-20 in 1× PBS at RT. Another round of 10% cleaving solution was incubated on the samples for 15 min at RT and washed. The RNAscope Fluoro probes T5–T6 were then incubated at 40 °C for 15 min to anneal to *Aspa* and *Vamp3* mRNA. The samples were then washed, mounted, and imaged as described for the first round of probes. RNAscope HiPlex Registration software (Advanced Cell Diagnostics) was used to align images between the first and second round of probe hybridization with DAPI images as reference. Background subtraction and quantification of signal intensity was done by ImageJ (NIH).

## Immunohistochemistry

Tissue for spinal cords, brains, or sciatic nerves were dissected and fixed in cold 4% PFA for 3 h on ice, rinsed in cold PBS, and immersed in 30% sucrose at 4 °C overnight. Samples were embedded in O.C.T. (Tissue-Tek) and sliced into 30 μm thin cross sections or 30 μm longitudinal sections for the spinal cord, 30 μm sagittal sections for brains using a cryostat (CM3050 S, Leica Microsystems). Tissue slices were air dried on Superfrost Plus slides (VWR) and frozen at −80 °C. After drying, borders were drawn between samples on the same slide with a Super PAP Pen (ThermoScientific 008899). For fluorescent staining, samples were permeabilized with PBST (0.1% Triton X-100 in 1 × PBS) for 5 min at RT, blocked with 10% donkey serum (abcam #ab7475) for goat-hosted primary antibodies and normal goat serum (Thermo-Scientific #PCN5600) for other antibodies in PBST at RT for 20 min, rinsed with 1% donkey or goat serum in PBST, and incubated in primary antibody solution (dilutions specified above) containing 1% goat or donkey serum at 4 °C overnight. The following day primary antibody was rinsed off with PBST for a 1 min wash, followed by three 20 min washes. Samples were then incubated with secondary antibody solutions in 1% goat or donkey serum in PBST for 2 h at RT. Excess secondary solution was washed off with one quick 1 min rinse in PBST, followed by three 20 min washes. Lastly, samples were washed with PBS and then mounted in Vectashield with DAPI (Vector, H-1200) with a coverslip.

## Imaging and quantification of immunohistochemistry on spinal cord cross sections

Fixed spinal cord cross sections were imaged using a Zeiss LSM 800 laser scanning confocal microscope with the Plan-Apochromat 10× objective. Images were acquired using the Zeiss Zen Blue software and quantified with the researcher blinded to the genotype. 4–5 animals per genotype were analyzed and the *N* number was specified in the Figures and Figure Legends.

For quantification of white matter area (MBP) and axonal abundance (NF200), the maximum intensity projection of a z-stack containing 6 slices with a z-step of 5 μm was analyzed using Fiji (ImageJ, NIH) through batch processing. First, the total area of the sample was measured through Percentile thresholding of the DAPI stain. Then, the area of MBP (2° rat AlexaFluor-594) or NF200 (2° mouse AlexaFluor-647) staining was determined by Otsu thresholding and divided by the total area.

For quantification of CC1, Olig2, NeuN, and GFP-positive cells, the maximum intensity projection of a z-stack containing 6 slices with a z-step of 5 μm was analyzed using Fiji through batch processing. The threshold was set at 2.5× the mean intensity, which was a stringent cutoff that excluded all signal from the secondary antibody staining alone: CC1 (2° mouse Alexa Fluor 594), Olig2 (2° rabbit Alexa Fluor 647), NeuN (2° mouse Alexa Fluor 594), and GFP (2° chicken Alexa Fluor 488). The ROIs for cells of each marker (total) were detected using the "Analyze Particles" function with a binary thresholding of 2.5× the mean intensity and an area range of 100–2000 μm. Then, then number of cells with signal in a different channel were measured by a binary thresholding of 2.5× the mean intensity and divided by the total.

## Electron microscopic analysis

Tissue samples for transmission electron microscopy (TEM) were prepared as previously described[35] and then submitted to the Stanford Cell Science Imaging Facility for serial dehydration, embedding, and sectioning. Images were acquired on a JEOL 1400 TEM.

Spinal cord and sciatic nerve samples were prepared as previously described[92]. Briefly, animals were anesthetized and perfused with PBS and 4% PFA. Tissues were immediately and carefully dissected out and post-fixed in cold KS-fixative at 4 °C, treated with 2% osmium tetroxide in cold Karlsson-Schultz fixative, serially dehydrated, and embedded in EmBed812 (EMS CAT#14120). Spinal cord samples were then sectioned at 1 μm and stained with Toluidine Blue to locate the dorsal column. 75–90 nm sections were acquired onto formvar/carbon-coated 50 mesh copper grids, stained for 30 s in 3% uranyl acetate in 50% acetone followed by staining for 3 min in 0.2% lead citrate. Images were acquired with a JEOL 1400 TEM.

All imaging and analysis were conducted blinded to the genotype. The myelination status of each axon was classified into one of three categories: (1) unmyelinated, (2) partially/fully ensheathed (defined as oligodendrocyte contact with at least half the circumference of an axon), and (3) wrapped/wrapping (defined as encircled by 2 or more compacted layers of electron-dense lines ["major dense lines"]). Approximately 500–800 axons from control samples and 900–1200 from iBot; Cnp-Cre samples were categorized over an area of 520 μm² per animal from 2 nonadjacent fields.

## Purification and culturing of cells

Primary oligodendrocyte precursors were purified by immunopanning from P5-P7 Sprague Dawley rat and P6–P7 transgenic mouse brains as previously described[38,93]. oligodendrocyte precursors were typically seeded at a density of 150,000–250,000 cells/10 cm dish and allowed to recover for 4 days in culture before lifting cells via trypsinization and distributing for transfection, proliferation, or differentiation assays. All plasticware for culturing oligodendrocyte precursors were coated with 0.01 mg/ml poly-D-lysine hydrobromide (PDL, Sigma P6407) resuspended in water. All glass coverslips for culturing oligodendrocyte precursors were coated with 0.01 mg/ml PDL, which was first resuspended at 100× in 150 mM boric acid pH 8.4 (PDL-borate).

To proliferate primary oligodendrocyte precursors, cells were cultured in serum-free defined media (DMEM-SATO base medium) supplemented with 4.2 μg/ml forskolin (Sigma-Aldrich, Cat#F6886), 10 ng/ml PDGF (Peprotech, Cat#100-13A), 10 ng/ml CNTF (Peprotech, Cat#450-02), and 1 ng/ml neurotrophin-3 (NT-3; Peprotech, Cat#450-03) at 37 °C with 10% CO₂. To induce differentiation, cells were switched to DMEM-SATO base media containing 4.2 μg/ml forskolin (Sigma-Aldrich, Cat#F6886), 10 ng/ml CNTF (Peprotech, Cat#450-02), 40 ng/ml thyroid hormone (T3; Sigma-Aldrich, Cat#T6397), and 1× N21-MAX (R&D Systems AR008).

## Cloning of pH-sensitive exocytosis reporters (VAMP2/3-pHluorin)

For live imaging of exocytosis in cultured oligodendrocyte precursors/oligodendrocytes, all reporter constructs contained super ecliptic pHluorin (referred to as pHluorin). Plasmids containing murine VAMP2-pHluorin and VAMP3-pHluorin were gifts from Dr. Stephanie Gupton (UNC Chapel Hill). VAMP2- and VAMP3-pHluorin were cloned into a pAAV vector backbone with a CMV promoter. The following plasmids were used to transfect rat oligodendrocyte precursors: CMV promoter-driven VAMP2-pHluorin, CMV promoter-driven VAMP3-pHluorin, and CMV promoter-driven mRuby3-caax (a membrane-bound marker).

## Cloning of dominant-negative VAMP proteins (dn-VAMP2/3)

The dn-VAMP2/3 constructs were assembled by InFusion cloning (Takara Bio) to anneal three DNA fragments in a pAAV vector backbone. The parent plasmid (pBZ-281), which contains the MBP promoter in a pAAV vector backbone, was linearized using the BamHI and BstEII restriction sites. The first fragment contained the cytosolic domain of either VAMP2 (aa 1–94) or VAMP3 (aa 1–81), and was cloned from the VAMP-pHluorin constructs provided by Dr. Stephanie Gupton. The second fragment contained the P2AT2A sequence, cloned from AddGene #87828[94], to encode a tandem self-cleaving peptide downstream of dn-VAMPs. The third fragment contained a GFP-caax cloned from pBZ-282. The final DNA plasmid contains MBP promoter – dnVAMP – P2AT2A – GFP-caax.

## Transfection of oligodendrocyte precursors

Proliferating rat oligodendrocyte precursors were lifted from tissue culture dishes, and centrifuged at 90 × g for 10 min. 250,000 oligodendrocyte precursors were gently resuspended into 20 μl of nucleofector solution (Lonza P3 Primary Cell 4D-Nucleofector V4XP-3032) with transfection-grade, endotoxin-free DNA prepared at 400 ng/μl using Qiagen Plasmid Plus Midi Kit (Qiagen 12945). For live-cell imaging of exocytosis, 300 ng of VAMP-pHluorin and 300 ng of mRuby-caax plasmids were co-transfected. For expression of dominant negative-VAMP proteins, 400 ng of GFP-caax, dn-VAMP2-P2AT2A-GFP-caax, or dn-VAMP3-P2AT2A-GFP-caax was used. In the co-transfection of dn-VAMP2 and dn-VAMP3, 200 ng of dn-VAMP2-P2AT2A-GFP-caax was combined with 200 ng of dn-VAMP3-P2AT2A-GFP-caax for 400 ng DNA total. Cells were then electroporated in a Lonza 4D-Nucleofector X Unit (AAF-1003X) assembled with a 4D-Nucleofector Core Unit (AAF-1002B) with pulse code DC-218. Electroporated cells rested for 10 min at RT before resuspension in antibiotic-free DMEM-SATO base media supplemented with proliferation or differentiation factors.

Each batch of 250,000 cells was distributed into 4 35 mm dishes with No. 1.5 glass coverslips (MatTek Corporation P35G-1.5-20-C) coated with PDL-borate for differentiation timepoints and technical replicates. Each dish was half-fed with freshly supplemented DMEM-SATO media every two days. For live-cell imaging of cells transfected with pHluorin, media was replaced with FluoroBrite DMEM-SATO (made with Fisher Scientific A1896701) supplemented with differentiation factors for 2 h at 37 °C, 10% CO₂ before imaging.

For imaging of cells transfected with dn-VAMP constructs, cell media was removed, washed with 1 × PBS, and fixed with 4% PFA for 15 min at RT. No further antibody staining was necessary to visualize GFP-caax. See "Image analysis of immunofluorescence of primary oligodendrocytes and transfected oligodendrocytes" below.

## Live-cell imaging of exocytosis in primary oligodendrocyte-lineage cells

Time-lapse imaging of exocytic events was performed on a Zeiss Axio Observer Z1 inverted microscope equipped with a Zeiss Axiocam 506 monochrome 6-megapixel camera, a stage top incubator (Okolab, H301-K-Frame) set to 37 °C, and a digital gas blender (Okolab, CO2-UNIT-3L) set to 10% CO₂ during image acquisition. Samples were imaged with a Plan-Apo 63×/1.40 Oil objective using widefield epi-fluorescence with a 12 V halogen lamp. Due to a high baseline level of VAMP-pHluorin intensity on the cell surface, cells were subjected to initial "pre-bleaching" consisting of 20 frames of 50 ms exposure at a 250 ms frame rate. Then, time-lapse sequences for exocytotic events were captured using an acquisition rate of 250 ms/frame for 1 min using the Zen Blue software. Images were viewed using Fiji/ImageJ software.

## Analysis of exocytotic events in primary oligodendrocyte cultures

Exocytotic events were quantified using a modified workflow adapted from the analysis pipeline published by the Gupton lab[25,26,95], which defines exocytotic events as non-motile Gaussian-shaped puncta with

transient intensity increases that reach four standard deviations above the local background intensity. The analysis was performed blinded to the timepoint of differentiation.

The total frequency and positions of exocytotic events within individual cells was extracted using published software, and then manually inspected. The soma for each cell was determined by thresholding the first frame of the pHluorin time-lapse to capture an ROI for the cell body, where baseline VAMP-pHluorin fluorescence was highest. All events with $x,y$ positions within the cell body ROI were designated as soma, while others were designated as processes/sheet.

The built-in parameters for predicting exocytotic fusion modes modeled on neuronal studies did not properly distinguish fusion modes for our events measured in oligodendrocytes. We instead assigned fusion modes in oligodendrocytes using the fluorescence decay of the event versus its bordering membrane, based on published observations that full-vesicle fusion events exhibit radial fluorescence spreading while kiss-and-run events do not. For each event, the intensity was determined for a circular ROI with a radius of 250 nm (intensity$^{center}$) and a radius of 500 nm (intensity$^{border+center}$) over time. We calculated the intensity of the bordering membrane as (intensity$^{border+center}$ – intensity$^{center}$). Then, we aligned the fluorescent time traces of each event by reassigning the maximum intensity$^{center}$ to $t_0$. We fit intensity over time to an exponential decay model using RStudio scripts developed in the Gupton lab to determine the half-life of fluorescence for time traces of intensity$^{center}$ ($t_{1/2}^{center}$) and of intensity$^{border}$ ($t_{1/2}^{border}$). We reasoned that a full vesicle fusion event with radial fluorescence spreading would exhibit a $t_{1/2}^{border}$ that is proportional to its $t_{1/2}^{center}$, while a kiss-and-run event would exhibit a rapid border decay with a $t_{1/2}^{border}$ that is relatively small. Therefore, we calculated a border decay ratio ($t_{1/2}^{border}/t_{1/2}^{center}$) for each event. The distribution of border decay ratios for all events clustered into two peaks, reflecting a population of events with radial fluorescence spreading centered around ($0.89 \pm 0.14$) and events without. Events with border decay ratios within 3 standard deviations of ($0.89 \pm 0.14$) were designated as full-vesicle fusion events, and events with ratios less than 3 standard deviations of ($0.89 \pm 0.14$) were designated as kiss-and-run.

### Live imaging of oligodendrocyte exocytosis in zebrafish

All zebrafish were maintained under standard conditions at the University of Edinburgh with approval from the UK Home Office according to its regulations.

To generate VAMP2/3-pHluorin constructs to image exocytosis in zebrafish, we first cloned zebrafish *vamp2/3* cDNA from a pool of total cDNA from 5dpf zebrafish of the AB strain, by carrying out high-fidelity PCR using Phusion DNA polymerase and primers 5′-AACCCGGTT-CAAAATGTCTGCC-3′ (vamp2 F, start codon underlined), 5′-CGCTTTAGGTGCTGAAGTACACAATG-3′ (vamp2 R, stop codon underlined), 5′-GTCCGCTCCAGGTGCAGATG-3′ (vamp3 F, start codon underlined, completed through subsequent cloning steps), 5′-CACAACCAGACTCTGTGCCACTT-3′ (vamp3 R, stop codon underlined). PCR products were TOPO-cloned (Zero Blunt™ TOPO™ PCR Cloning Kit, Thermo Fisher Scientific) into pCRII vector backbones. We then recombined vamp2/3 cDNA including a Kozak sequence upstream of the start codon, and excluding the stop codons, into Gateway-compatible middle-entry vectors to use with the tol2kit. To do this, we used Phusion and primers 5′-GGGGACAAGTTTGTA-CAAAAAAGCAGGCTGCCACCATGTCTGCCCCAGCCGGAGC-3′ (attB1-vamp2, Kozak and start codon underlined), 5′-GGGGACCACTTTGTA-CAAGAAAGCTGGGTGGTGCTGAAGTACACAATGATTATAATG−3′ (attB2R-vamp2-nostop, cDNA underlined), 5′-GGGGACAAGTTTGTAC-AAAAAAGCAGGCTGCCACCATGTCCGCTCCAGGTGCAG-3′ (attB1-vamp3, Kozak and start codon underlined), and 5′-GGGGACCACTT-TGTACAAGAAAGCTGGGTTGACTGCGACCAGATGACAATGATG−3′ (attB2R-vamp3-nostop, cDNA underlined) to amplify attB-vamp2/3 PCR products from the pCRII vectors, and recombined these with

pDONR221 using BP Clonase II to generate pME-VAMP2/3-nostop. We then generated a Gateway-compatible 3′-entry vector containing super ecliptic pHluorin, using pcDNA3-SypHluorin4x (Addgene #37005)[96] as a template for PCR and primers 5′-GGGGACAGCTTTCTTGTA-CAAAGTGGTCCCCCATGGATCTAGCCACC**ATG**−3′ (attB2-(linker) pHluorin F, linker region underlined and pHluorin coding sequence in bold) and 5′-GGGGACAACTTTGTATAATAAAGTTGT**TTA**CGATAAGC TTGATCGAGCTCCA−3′ (attB3R-pHluorin R, terminal linker underlined and stop codon in bold) to amplify an attB-containing product, and recombining it with pDONRP2RP3 using BP clonase II. This plasmid maintains the reading frame when recombined with the coding sequences in pME-vamp2/3. All entry vectors were verified by Sanger sequencing. To generate final Tol2 expression vectors, a 5′-entry plasmid containing 10 upstream activating sequence (UAS) repeats (plasmid #327 from the tol2kit), the vamp2/3 middle-entry plasmids and the 3′-pHluorin plasmids were recombined with a destination vector pDestTol2pA2 from the tol2kit in a LR reaction using LR Clonase II Plus. All expression vectors were verified by diagnostic restriction digest and colony PCR.

To express the vectors 10UAS:vamp2/3-pHluorin (VAMP2/3-pHluorin) in individual oligodendrocyte-lineage cells, we injected 1–5 pg of vector DNA with 25 pg of tol2 transposase mRNA into one-cell stage eggs of the transgenic lines Tg(olig1:KalTA4)[97] or Tg(claudinK:Gal4)[98]. This approach leads to sparse expression in olig1+ cells (OPC/premyelinating stage) or claudinK+ cells (early myelinating/mature OL stage) since the transcription factors KalTA4/Gal4 specifically recognize the UAS repeats and drive expression of the downstream genes in a mosaic manner. The morphology of each cell was used to distinguish between stages: OPCs/premyelinating OLs had many ramified processes and no myelin sheaths; early OLs had many (average 25) but short sheaths (5–11 μm); mature OLs had on average 11–12 sheaths of longer length (19–34 μm).

For live-imaging of VAMP2/3-pHluorin, 4–5dpf larvae were first pre-screened for isolated cells of interest (typically only one cell per animal was imaged) and then anesthetized with tricaine and mounted on their sides in a drop of 1.5% low-melting point agarose on a glass-bottom petri dish. Larvae were imaged with a Zeiss LSM880 confocal with Airyscan in Fast mode and a Zeiss W Plan-Apochromat 20×/1.0 NA water-dipping objective, and a 488 nm laser. We used 3–6.5X zoom and acquired 500–1500 (X) × 110–390 (Y) pixels (6–15 pixels/μm) in a small z-stack (7–16 z-slices, 1.2–2.1 μm z-step), sampling the entire cell of interest repeatedly at a frequency of 0.6–2 Hz (mean 1 Hz), for 8–26 min. To increase the signal-to-noise ratio of pHluorin-reported exocytosis events, we bleached the baseline level of VAMP2/3-pHluorin at the cell's membrane by setting the laser power to 100% during the first ~30–45 s of acquisition (we excluded the bleaching period from image analysis).

For image analysis, we used Fiji/ImageJ[99] and Python scripts for most processing. Time-lapses were first pre-processed by bleach-correction with exponential curve fitting and registration, where needed, using the TurboReg plugin[100] with rigid transformation. Putative pHluorin events were identified in the pre-processed time-lapse, aligned to a $\Delta F/F_{avg}$ timelapse (proportional change over the all-time average intensity) to aid discrimination of the event start. For each potential event, we defined its region of interest using Fiji's Wand tool to select the maximum intensity pixel and connected region over a third of the maximum fluorescence intensity. We used this ROI to determine the increase in fluorescence intensity relative to the baseline (F0), defined as the mean intensity in the four frames preceding the event, and only considered events in which fluorescence increased at least to four standard deviations above the baseline, persisting in at least 3 consecutive frames (~3 s). We excluded from further analysis events that showed significant motion throughout their duration (but included events with limited displacement). Collection of these parameters was automated using

custom written ImageJ macros, but all events were manually inspected.

## Immunofluorescence of primary mouse oligodendrocytes

Primary oligodendrocyte precursors from iBot;*Cnp*-Cre and control littermate mice were harvested and cultured as described above in "Purification and Culturing of Cells". Cells were seeded onto 12 mm glass coverslips (Carolina Biological Supply No. 63–3029) at a density of 10,000 cells/coverslip in differentiation media.

At the specified day of differentiation, cell media was removed and replaced with one wash of 1 × PBS. Cells were then treated with 4% PFA for 15 min at RT, followed by three washes with 1xPBS and permeabilization in 0.1% Triton X-100 in PBS for 3 min at RT. Prior to staining, cells were incubated in a blocking solution of 3% BSA in PBS for 20 min at RT. Then, primary antibodies (mouse anti-GalCer hybridoma and rat anti-MBP) were added in a 3% BSA solution for overnight incubation at 4 °C. On the following day, the primary antibody solution was rinsed off with three washes of PBS, and then incubated with secondary antibodies (anti-mouse AlexaFluor 594 and anti-rat AlexaFluor 647) in 3% BSA for 1 h at RT. After one wash with PBS, CellMask Blue stain (1:1000) was incubated to stain all cells for 10 min at RT, followed by three additional rounds of washing with PBS. Stained cells were mounted onto microscope slides (Fisher Scientific 12-550-143) in Fluoromount G (SouthernBioTech, 0100-20).

Cells were imaged by widefield epifluorescence with a Zeiss Axio Observer Z1 using the Plan-Apo 20×/0.8 NA objective for membrane area quantification and using the Plan-Apo 63×/1.40 Oil objective for visualizing vesicle accumulation in cultured cells. Images were acquired blinded to the genotype with identical illumination and acquisition conditions per biologically independent replicate.

## Image analysis of immunofluorescence of primary oligodendrocytes and transfected oligodendrocytes

Images were analyzed through batch processing in Fiji/Image J[99]. For primary oligodendrocytes, data from 2–4 replicate coverslips per biologically independent replicate were averaged. Cells from 5 biologically independent replicates ($n = 5$ pairs of control and iBot;*Cnp*-Cre pups) were analyzed.

To quantify the percentage of cells expressing a specific marker (Supplementary Fig. 6a–d), the CellMask Blue image was first thresholded against unstained samples to create cell body ROIs for the total number of cells per image. Then, the GFP, GalCer, and MBP images were separately thresholded against their respective secondary antibody controls to make binary masks. The number of cell body ROIs with positive signal in each channel was quantified.

To quantify the membrane area of primary oligodendrocytes (Fig. 3c), cells were manually segmented and thresholded in the GalCer channel to create a refined ROI selection that closely outlined the membrane border of each cell. The area of the refined ROI was then measured (See Supplementary Fig. 6e).

To quantify the membrane area and mean expression of dn-VAMP-transfected oligodendrocytes (Supplementary Fig. 9b–d), cells were manually segmented and thresholded in the GFP channel to create a refined ROI selection that closely outlined the membrane border of each cell for the area and the mean GFP intensity to compare expression levels.

## Measuring cell survival

Primary oligodendrocyte precursors from iBot;*Cnp*-Cre and control littermate mice were harvested and cultured as described above in "Purification and Culturing of Cells". Cells were re-seeded onto 24-well plates (Fisher Scientific 08-772-1) at a density of 5000 cells/well in media with proliferation factors. After 48 h of recovery, fresh media with proliferation and differentiation factors was added, and the plates were transferred to an IncuCyte ZOOM live cell imaging system (Essen

Bioscience) set at 37 °C, 10% $CO_2$, and imaged every 12 h by phase contrast with the 20× objective in 9 different regions per well. Half the media was replaced in each well every 48 h. After 5 days, ethidium homodimer-1 (EthD-1, Thermo Fisher Scientific E1169) was added to the wells at a 1:1000 dilution and incubated for 30 min before imaging for phase contrast, green, and red fluorescence.

Images were analyzed blinded to the genotype through batch processing in Fiji/Image J. Phase contrast images were inverted and thresholded to create ROIs using the "Analyze Particles" function with an area range of 100–5000 µm² and a circularity of 0.05–1.00, recording the number of total cells per image. Images for red fluorescence were thresholded against unstained cells, and the intensity above threshold for each ROI in the red channel was measured. ROIs with positive intensity in the red channel were counted as dead cells, and the number of live cells was calculated by subtracting the dead cells from the total. 9 images per well were used to calculate the total and percentage for each biological replicate.

## Myelinating co-cultures with RGCs

Our protocol for myelinating co-cultures with CNS-derived axons was adapted from previous publications[39,40] with minor modifications. Dissection and dissociation of retina, immunopanning for retinal ganglion cells (RGCs) with anti-Thy 1.1, and media composition with growth factor supplementation was performed as previously published.

To create re-aggregates, we seeded freshly harvested RGCs into UV-sterilized PCR tubes (Fisherbrand 14-230-215) at a density of 10,000 cells/100 µl of media. After 24 h of recovery, we half-fed each tube by exchanging 50 µl of media with freshly supplemented growth factors. After 48 h of recovery, we transferred the 100 µl-cell suspension containing "re-aggregated" clumps of RGCs from each tube onto a PDL-borate coated 12 mm glass coverslip situated in a 24-well plate. Each well was half-fed with fresh growth factors every 72–96 h. After 10–14 days, the RGC re-aggregates formed dense beds of radially protruding axons.

Mouse oligodendrocyte precursors freshly harvested from P5 to P7 brains were seeded directly onto RGCs at a density of 40,000–50,000 cells/well. After 24 h, 1 µm of the gamma-secretase inhibitor DAPT (Calbiochem Cat. No. 565784) was added to each well to promote ensheathment. The co-cultures were fed every 48–72 h and fixed after 7 days for immunofluorescence as previously described in published protocols. Fixed co-cultures were stained with rat anti-MBP and mouse anti-NF200, followed by anti-rat AlexaFluor 594 and anti-mouse AlexaFluor 647.

### Image acquisition and sheath analysis for neuron-oligodendrocyte co-cultures.

Neuron-oligodendrocyte co-cultures were imaged by widefield epifluorescence with a Zeiss Axio Observer Z1 using the Plan-Apo 20×/0.8 NA objective using 3×3 tiling with 10% overlap on ZenBlue software. Images were acquired blinded to the genotype of the oligodendrocytes. Regions with dense axon signal (NF200 staining) adjacent to the re-aggregated cell bodies were chosen. For all images acquired, the axon density ranged from 6-7% of the total field of view (Supplementary Fig. 8c).

Images were analyzed blinded to the genotype through batch processing in Fiji/Image J. Cells from 4 biologically independent replicates ($n = 4$ pairs of control and iBot;*Cnp*-Cre pups) were analyzed. First, individual oligodendrocytes were manually segmented in the MBP channel. For each segmented region, linear MBP tracks were manually traced and saved as line ROIs (see Supplementary Fig. 8). The number of sheaths was quantified as the number of line ROIs per oligodendrocyte, and the length of sheaths was measured as the lengths of the line ROIs. For regions where sheaths of two oligodendrocyte cell bodies overlapped, the total number of sheaths were divided by two to obtain an average sheath number per cell body. The quantified number of sheaths per cell may be underestimated in the control samples due to the limited resolution of clustered sheaths. In each biologically

independent replicate, 34–48 individual oligodendrocytes were analyzed.

## Quantification of sheath length in deep cortical layers

Fixed sagittal brain slices were imaged using a Zeiss LSM 800 laser scanning confocal as described in "Imaging and quantification of immunohistochemical staining". For sheath length analysis, we focused on the deep cortical layers above the cingulate cortex, where myelination is sparse during the second postnatal week[41]. Discrete, linear MBP signal were manually traced blinded to the genotype of the sample. For each biologically independent replicate, 176–440 individual sheaths were traced in Fiji/ImageJ and analyzed.

## Quantification of myelin coverage from IHC internode analysis and electron microscopy sections

Transmission EM (TEM) micrographs reveal only an ultrathin (~75–90 nm) section of tissue, while the length of myelin sheaths are much larger (on the order of 100 μm). Hypomyelination (lower percentage of myelinated axons per section) could be due to a bona fide ensheathment defect, or instead due to a combination of fewer oligodendrocytes, reduced myelin sheath length, and/or reduced sheath number per oligodendrocyte. To distinguish between these possibilities, we compared estimated values for myelin coverage from our immunostaining analysis (Supplementary Fig. 2, Fig. 4) to our TEM quantification of myelination (Fig. 1g).

Myelin coverage (percent of total axonal surface myelinated) in a region of white matter can be approximated from the product of total number of oligodendrocytes (N) times the average length of myelin sheaths (L) times the average number of sheaths per oligodendrocyte, assuming the number of axons is the same:

$$\text{Approximated myelin coverage} = N \cdot L \cdot I \quad (1)$$

The relative amount of myelin coverage between iBot and wild-type (WT) mice can be expressed as the ratio of estimated myelin coverage between the two genotypes:

$$\text{Relative myelin coverage (iBot to WT)} = \frac{N(iBot) \bullet L(iBot) \bullet I(iBot)}{N(WT) \bullet L(WT) \bullet I(WT)} \quad (2)$$

To determine whether the hypomyelination seen in iBot mice by EM can be explained entirely by reduced myelin coverage (primarily due to shorter sheaths), we calculated the expected relative myelin coverage of both iBot and WT mice using numbers measured in the following experiments:

Number of oligodendrocytes, **N:** CC1+ cells in dorsal spinal cord white matter, Supplementary Fig. 2f.

Average length of myelin sheaths (i.e., internodes), **L:** In vivo sheath length measurements, Fig. 4g. (Note that myelin was too dense in WT spinal cord tissue sections to accurately measure the length of individual sheaths, so we used available measurements from the deep cortical layers that are still sparsely myelinated at P12 when samples were acquired. Although myelin sheaths are on average shorter in the cortex than in the spinal cord[101,102], our estimations rely on the ratio of iBot to control rather than exact lengths.)

Average number of myelin sheaths per oligodendrocyte, **I:** Myelinating co-culture measurements, Fig. 4c. (Note, as above, due to

myelin density in the WT spinal cord, we were able to make the most precise measurements of sheath number per oligodendrocyte from our co-culture experiments.)

Plugging in these values to Formula 2 (see calculation in Table 1):

$$\text{Relative myelin coverage by IHC analysis(iBot to WT)} = \frac{N(iBot) \bullet L(iBot) \bullet I(iBot)}{N(WT) \bullet L(WT) \bullet I(WT)} = \mathbf{0.399} \quad (3)$$

How closely does this estimate match the observed myelin coverage from spinal cord TEM images? We calculated the total number of axons myelinated (ensheathed plus wrapping/wrapped) in the dorsal white matter in both iBot and WT littermate mice (see calculation in Table 2):

$$\text{Relative myelin coverage by TEM (iBot to WT)} = \frac{\text{number of axons myelinated (iBot)}}{\text{number of axons myelinated (WT)}} = \mathbf{0.401} \quad (4)$$

Therefore, the hypomyelination seen in iBot spinal cords by TEM (~40% coverage compared to WT) can be explained due to the significantly shorter sheaths made by iBot oligodendrocytes and small (albeit not reaching significance) reductions in number of oligodendrocytes and number of sheaths per oligodendrocyte (see Supplementary Fig. 13a).

## AAV-mediated sparse labeling of oligodendrocytes in mouse spinal cord

Serotype AAV-DJ was prepared by the Stanford Neuroscience Gene Vector and Virus Core, and stored at −80 °C until use. Wild-type C57BL6 neonatal mouse pups were injected within 6–24 h after birth. Each pup was wrapped in a Kimwipe and incubated on ice for 4–5 min until the pup was unresponsive to a toe-pinch. Cryoanesthetized pups were injected using a 10 μl Hamilton syringe (Model 80308 701SN, Point Style 4, 32 gauge, 20 mm length, and 12°). 1 μl of AAV (diluted to $1.0 \times 10^{13}$ viral genome/ml in PBS) was mixed with 0.5 μl of Trypan blue and slowly injected into the lumbar spinal cord, which appeared as a blue stripe that colors the spinal cord. The pups were incubated on a heating bench set at 34–37 °C for 15 min until they were able to move, and then returned to their home cage.

Spinal cords were extracted at P12 as detailed in the "Immunohistochemistry" section. 30 μm-thick longitudinal sections were mounted onto Superfrost Plus slides (VWR) in Vectashield with DAPI (Vector, H-1200) with a coverslip. No antibody staining was necessary to visualize the GFP fluorescence. Tissue samples were imaged blinded to the virus condition using the Zeiss LSM 880 with the Plan-Apochromat 20× objective as a z-stack containing 6 slices with a z-step of 5 μm. The maximum intensity projection of each stack was analyzed using Fiji by manually tracing discrete sheaths blinded to the virus condition.

## Confocal imaging and analysis of myelin bulges

For high-resolution comparison of myelin morphology, the cingulate cortex of MBP-stained sagittal brain slices were imaged blinded to the genotype using the Zeiss LSM 800 laser scanning confocal microscope with a Plan-Apo 63×/1.4 NA oil objective. Z-stacks with 25 μm range and 2.5 μm z-steps were acquired using identical settings for the frame size

**Table 1 | Calculation for relative myelin coverage by internode analysis (iBot to WT)**

| | N (# of oligodendrocytes, from spinal cord CC1 staining) | L (avg. internode length, in vivo from cortex) | I (avg. internode #), from co-cultures | Product (Estimated myelin coverage) | Ratio (iBot to control) |
|---|---|---|---|---|---|
| iBot | 1558 | 39.6 | 6.85 | 422,623.1 | **0.399** |
| Control | 1999 | 54.2 | 9.78 | 1,059,621.9 | |

**Table 2 | Calculation for relative myelination by cross-sectional TEM (iBot to WT)**

| | Number of axons myelinated per 200 µm² (ensheathed + wrapped/wrapping) | Ratio (iBot to control) |
|---|---|---|
| iBot | 28.4 | **0.401** |
| Control | 70.9 | |

and scan speed with no averaging between samples and biologically independent replicates.

To quantify the regional diameter of myelin sheaths, maximum intensity projections were thresholded to preserve gray-level moments ("Moments"), which highlighted discrete regions of high MBP intensity within each sample. Then, lines perpendicular to the longest width of discrete MBP regions were manually drawn and saved as ROIs. Line scans for each ROIs were processed in batch to identify the distance between local intensity maxima, which represented the distance between two linear tracks of MBP signal. Heterogeneity in MBP staining was depicted in BatlowK, which was developed as a perceptually uniform, universally-readable look up table.

For the volume reconstruction of a myelin bulge, a 5 to 7 µm z-stack of an MBP-stained spinal cord cross section from a control or iBot;*Cnp*-Cre P12 mouse was acquired using the Zeiss LSM 800 laser scanning confocal microscope with a Plan-Apo 63×/1.4 NA oil objective with the 32-channel Airyscan detector. The frame size (1952×1952 pixels), scan speed, and z-step (170 nm) were optimized by the Zeiss ZenBlue software to satisfy the Nyquist criteria for 3D imaging. The acquired z-stack was then deconvoluted using the default Wiener filter settings in Airyscan processing from the Zeiss ZenBlue software. The deconvoluted z-stack was then reconstructed and visualized using the Volume Viewer plugin from Fiji/ImageJ.

### Surface biotinylation and immunoprecipitation

Our protocol for surface biotinylation of primary oligodendrocytes was adapted from a previously published protocol[103]. $1 \times 10^6$ primary oligodendrocyte precursors were replated onto 10 cm dishes directly into differentiation media. Two 10 cm dishes were grown for each condition in each biologically independent replicate. After 5 days of differentiation in culture, cells were washed with cold PBS (10 ml × 3 rinses on the plate) and then incubated with 1 mg of Sulfo-NHS-SS-biotin (BioVision #2323) dissolved in 10 ml of cold PBS on a rotating platform at 4 °C. Non-biotinylated samples were incubated with cold PBS only. After 1 hr, excess biotinylation reagent was quenched with 65 mM Tris pH 7.5 at 4 °C, 150 mM NaCl, 1 mM $MgSO_4$, 1 mM $CaCl_2$ (10 ml for 5 min × 3) followed by a rinse with cold PBS. Cells were harvested by cell scraping into 2 ml of PBS with a protease inhibitor cocktail (Roche cOmplete 4693159001). The cell suspension was centrifuge at $200 \times g$ for 10 min, and the supernatant was discarded to isolate the pellet. The cell pellet was flash-frozen in liquid nitrogen for storage at −80 °C.

For immunoprecipitation, the cell pellets were resuspended in 500 µl of cold lysis buffer (150 mM NaCl, 0.5% SDS, 50 mM Tris, pH 8.0) supplemented with Roche protease inhibitor cocktail. Cells were lysed by mechanical shearing through a 27.5 G needle with 15 passes, on ice. Unlysed cells were cleared via centrifuging at $3000 \times g$ for 10 min at 4 °C. Meanwhile, streptavidin magnetic beads (Pierce 88817, 25 µl of beads per sample) were washed with three rounds of 500 µl of lysis buffer. Then, cleared cell lysate was incubated with the streptavidin beads on a rotating platform overnight in the cold room. After removing the flow-through, the beads are washed with 2 × 1 ml of lysis buffer for 10 min, followed by 1 M KCl for 10 min, and another 2 × 1 ml lysis buffer for 10 min. Beads were finally washed with cold PBS before mass spectrometry analysis.

10 µl of the cleared cell lysate was loaded onto a 4–12% Bis-Tris pre-cast gel (ThermoFisher NW04120BOX) and transferred onto a nitrocellulose membrane using the iBlot 2 dry blotting system (ThermoFisher). The membrane was then blocked in 3% BSA in 1xPBST (PBS with 0.1% Tween-20) for 1 h at RT, and then incubated with a streptavidin-horseradish peroxidase (HRP) conjugate (1:10000, ThermoFisher S-911) to visualize biotinylated proteins and anti-GFP conjugated to Alexa Fluor 488 (1:1000, Santa Cruz sc-9996 AF488) in 3% BSA 1 × PBST overnight at 4 °C. The membrane was washed 3 × 10 min at RT before incubating with the HRP chemiluminescent substrate (SuperSignal West Dura 34075) and imaging for chemiluminescence and fluorescence in the 488 nm channel.

### Mass spectrometry and analysis

In a typical mass spectrometry experiment, beads were reconstituted in TEAB prior to reduction in 10 mM DTT followed by alkylation using 30 mM acrylamide to cap cysteine residues. Digestion was performed using Trypsin/LysC (Promega) in the presence of 0.02% ProteaseMax (Promega) overnight. Following digestion and quenching, eluted peptides were desalted, dried, and reconstituted in 2% aqueous acetonitrile prior to analysis. Mass spectrometry experiments were performed using liquid chromatography (LC/MS) using an Acquity M-Class UPLC (Waters) followed by mass spectrometry using an Orbitrap Q Exactive HF-X (Thermo Scientific). For a typical LC/MS experiment, a flow rate of 300 nL/min was used, where mobile phase A was 0.2% (v/v) formic acid in water and mobile phase B was 0.2% (v/v) formic acid in acetonitrile. Analytical columns were prepared in-house by pulling and packing fused silica with an internal diameter of 100 microns. These columns were packed with NanoLCMS solutions 1.9 µm C18 stationary phase to a length of approximately 25 cm. Peptides were directly injected into the analytical column using a gradient (2 to 45% B, followed by a high-B wash) of 90 min. Mass spectrometry was operated in a data-dependent fashion using Higher Energy Collison Dissociation (HCD) for peptide fragmentation on the HF-X.

For data analysis, the .RAW data files were checked using Preview (Protein Metrics) to verify calibration and quality metrics. Data were processed using Byonic v4.1.5 (Protein Metrics) to identify peptides and infer proteins using the isoform-containing *Mus musculus* database from Uniprot, concatenated with common contaminant proteins, e.g., human keratins. Proteolysis with Trypsin/LysC was assumed to be semi-specific allowing for ragged n-termini with up to two missed cleavage sites, and allowing for common modifications. Precursor and fragment mass accuracies were held within 12 ppm. Proteins were held to a false discovery rate of 1%, using standard approaches described previously[104].

The label-free peptide counts from all runs were compiled and further analyzed using Perseus (version 1.6.5.0)[105]. The peptide counts were $\log_2$ transformed with missing values imputed from the normal distribution to calculate $\log_2$ fold change between control ($n = 2$) and iBot;*Cnp*-Cre ($n = 3$) samples. Statistical significance of fold changes was assessed by an unpaired Student's t-test with two-tailed distribution. This analysis workflow was repeated for biotinylated control versus non-biotinylated control samples.

Non-specific interactions were filtered away as proteins with less than a 2-fold difference between biotinylated and non-biotinylated control samples. For the remaining surface biotinylation-enriched proteins, the $\log_2$ fold change between control and iBot;*Cnp*-Cre with false discovery rate-adjusted p-values were plotted in Fig. 6b. VAMP2/3-dependent hits were defined as proteins that were at least 2-fold depleted from iBot;*Cnp*-Cre with an adjusted *p*-value < 0.05.

A functional gene enrichment analysis was performed with the VAMP2/3-dependent hits listed as an ordered query on the g:Profiler web server[106] for GO cellular components with a term size cutoff at 1000. Protein interaction networks were extracted from the STRING database and visualized on Cytoscape 3.9.0 software.

Article

## Surface staining of primary oligodendrocytes

Primary oligodendrocyte precursors from iBot;*Cnp*-Cre and control littermate mice were harvested and cultured as described above in "Purification and Culturing of Cells". Cells were seeded onto 12 mm glass coverslips (Carolina Biological Supply No. 63-3029) at a density of 10,000 cells/coverslip in differentiation media.

After 5 days in differentiation media, cell media was removed and replaced with one wash of 1 × PBS. Cells were then treated with 4% PFA for 15 min at RT, followed by three washes with 1 × PBS with no permeabilization. Cells were blocked with 3% BSA in PBS for 20 min at RT. Then, anti-MAG clone 513 (1:20, EMD Millipore MAB1567) in 3% BSA was incubated overnight at 4 °C to mark surface-exposed MAG. On the following day, the MAG antibody solution was washed with three rinses of PBS, and then incubated with the secondary anti-mouse IgG-Alexa Fluor 594 conjugate (Fisher Scientific A21203) in 3% BSA for 1 h at RT. Excess secondary was washed, and the surface MAG-bound secondary was fixed by another round of 4% PFA for 10 min at RT.

Following three PBS rinses to remove PFA, the cells were permeabilized with 0.1% TritonX-100 to reveal intracellular MAG. Then, the cells were re-incubated with anti-MAG clone 513 (1:100, EMD Millipore MAB1567) in 3% BSA overnight at 4 °C to mark intracellular MAG. The next day, excess MAG antibody solution was washed with three rinses of PBS, and then incubated with the secondary anti-mouse IgG-Alexa Fluor 647 conjugate (Fisher Scientific A31571) in 3% BSA for 1 h at RT. After one wash with PBS, CellMask Blue stain (1:1000) was incubated to stain all cells for 10 min at RT, followed by three additional rounds of washing with PBS. Stained cells were mounted onto microscope slides (Fisher Scientific 12-550-143) in Fluoromount G (SouthernBioTech, 0100-20). Cells were visualized by widefield epifluorescence with a Zeiss Axio Observer Z1 using the Plan-Apo 20x/0.8 NA objective. Cells were imaged blinded to the genotype with identical illumination and acquisition conditions.

To quantify the ratio of surface-to-total MAG intensity, cells were manually segmented on the CellMask Blue channel using the freehand tool and thresholded in the 647 nm channel (total MAG) to create a refined ROI that outlines the cell. For each ROI, the integrated intensity was measured in each channel for surface-exposed MAG in the 594 nm channel (intensity$^{surface}$) and total MAG in the 647 nm channel (intensity$^{total}$).

To correct for baseline differences in the intensity values of each channel, the mean background intensity of 4 non-overlapping 25,000 μm$^2$ regions with no cells was measured. For each cell, the mean background was multiplied by the ROI area to obtain the background intensity, which was subtracted from the integrated intensity in each channel, i.e., (intensity$^{surface}$) – (area$^{cell}$*background$^{594}$). These corrected integrated intensity values was used to obtain a ratio for surface-to-total MAG.

## Imaging and analysis of nodes of Ranvier

Longitudinal sections of the lumbar spinal cord were imaged using the Zeiss LSM 880 with the Plan-Apochromat 20× objective as a z-stack containing 5 slices with a z-step of 1 μm. The maximum intensity projection of each stack was analyzed using Fiji through batch processing. First, separate binary images for Caspr staining (2° rabbit AlexaFluor-647) and for AnkG staining (2° mouse AlexaFluor-594) were created by thresholding by preserving gray-level moments ("Moments"). Then the binary images were overlaid to obtain elliptical ROIs for the combined paranodes and node. Line ROIs were generated to span the bounding rectangle of each elliptical ROI and cross through the centroid. Intensity line scans of Caspr and AnkG were used to determine the number of local maxima present within each node ROI. Mature nodes were defined as ROIs with two local maxima for Caspr and one maxima for AnkG. Heminodes were defined as ROIs with one local maximum for both Caspr and AnkG. Clusters were defined as ROIs with one maximum in either Caspr or AnkG.

## Data analysis and statistics

All analysis was conducted blinded to the genotype. Data analysis and statistics were done using Microscoft Excel 16.63.1 and GraphPad Prism 9.0 software. Heat map representations of exocytosis events were produced using Matlab R2021b and RStudio 2021.09.0. Descriptive statistics (mean, standard error of mean, and N) were reported in Figure Legends or Supplementary Tables. Statistical tests are stated in the Figure Legends. For cellular assays, the biological replicate represented the mean of technical replicates from one mouse brain. In figures with single cell analysis, we represented the distribution of data in the SuperPlots format to depict cell-level and experimental variability[107].

## Reporting summary

Further information on research design is available in the Nature Research Reporting Summary linked to this article.

## Data availability

The data generated in this study are provided in the Supplementary Information and Source Data files. Source data are provided with this paper. The mass spectrometry proteomics data have been deposited to the ProteomeXchange[108] Consortium via the PRIDE[109] partner repository with the dataset identifier PXD036174 and 10.6019/PXD036174. Step-by-step protocols are available from the corresponding or first authors upon request. All DNA constructs created in this study have been deposited at Addgene: AAV_pCMV-Vamp2-pHluorin (Addgene plasmid 190151), AAV_pCMV-Vamp3-pHluorin (Addgene plasmid 190152), AAV_pMBP-dnVamp2-P2AT2A-EGFP-caax (Addgene plasmid 190153), AAV_pMBP-dnVamp3-P2AT2A-EGFP-caax (Addgene plasmid 190154), AAV_pMBP-EGFP-caax (Addgene plasmid 190155). Correspondence and requests for all other materials should be addressed to M.L. or J.B.Z. Source data are provided with this paper.

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

## Acknowledgements

We thank current and past members of the Zuchero lab (especially Alexandra Munch, Miguel A. Garcia, Graham Jones); Akiko Nishiyama, Ye Zhang, Christopher Fekete, and Lin Pan for their discussion and support; Jonah Chan, Suzanne Pfeffer, and Mark von Zastrow for valuable input at the beginning of the project; Klaus Nave for kindly sharing the *Cnp*-Cre mouse line; Frank Pfrieger and Shane Hentges for the iBot mouse line; and Stephanie Gupton for the VAMP-pHluorin plasmids. Images and diagrams were created using BioRender.com. We gratefully acknowledge the Stanford University Cell Sciences Imaging Core Facility for EM data collection, especially John Perrino and Ibanri Phanwar for processing and staining EM samples (RRID:SCR_017787: supported by ARRA Award Number 1S10RR026780-01 from the National Center for Research Resources [NCRR]. Its contents are solely the responsibility of the authors and do not necessarily represent the official views of the NCRR or the National Institutes of Health). We thank Ryan Leib, Fang Liu, Norah Brown, and Kratika Singhal at the Vincent Coates Foundation Mass Spectrometry Laboratory, Stanford University Mass Spectrometry for data collection and input on analysis (supported in part by NIH P30 CA124435 utilizing the Stanford Mass Spectrometry Shared Resource RRID:SCR_017801). We also thank the Stanford Neuroscience Gene Vector and Virus Core for producing the adeno-associated viruses (pMBP-GFP-caax and pMBP-dn-VAMP2-P2AT2A-GFP-caax) used in this study. This project was supported by the NEI T32 Vision Research Training Grant [T32EY027816] (M.L.), Toray Industries, Inc. (K.T.), Chancellor's Fellowship from the University of Edinburgh (R.G.A.), Stanford Medical Scientist Training Program [T32 GM007365-45] and Stanford Bio-X Interdisciplinary Graduate Fellowship (M.H.C.), Stanford Graduate Fellowship (M.I.), the McKnight Endowment Fund for Neuroscience (J.B.Z.), the Stanford Bio-X Interdisciplinary Initiatives Seed Grants Program (IIP) [R9-24] (J.B.Z.), the National Multiple Sclerosis Society Harry Weaver Neuroscience Scholar Award (J.B.Z.), the Beckman Young Investigator Award (J.B.Z.), the Myra Reinhard Family Foundation (J.B.Z.), the National Institutes of Health R01NS119823 (J.B.Z.), and the Koret Family Foundation (J.B.Z.). M.L. is a Merck-sponsored fellow of the Helen Hay Whitney Foundation and a Wu Tsai Neurosciences Interdisciplinary Scholar.

## Author contributions

J.B.Z., K.T., and M.L. conceived of the project. M.L. and K.T. designed, performed, and analyzed all experiments, with the following exceptions. R.G.A. designed, performed, and analyzed the live imaging in zebrafish in Fig. 2d–f. M.H.C. produced the reagents and optimized the protocol for the co-culture experiments in Fig. 4a–d. K.W. harvested the sciatic nerves in Supplementary Fig. 4. M.I. optimized the transfection protocol and initial rounds of pHluorin imaging for Fig. 2a–c. H.K. optimized the methodology for AAV-mediated sparse labeling of oligodendrocytes in the spinal cord for Supplementary Fig. 9f. M.L., K.T., and J.B.Z. wrote the manuscript. All authors reviewed and gave feedback on the manuscript.

## Competing interests

K.T. is an employee of Toray Industries, Inc. The remaining authors declare no competing interests.
