## [Peer Review File · Nature Communications]

CNS myelination requires VAMP2/3-mediated membrane expansion in oligodendrocytesEditorial Note: Parts of this Peer Review File have been redacted as indicated to maintain the confidentiality of unpublished data.

REVIEWER COMMENTS

Reviewer #1 (Remarks to the Author):

This review was completed with a graduate student as a training exercise. We read the manuscript and prepared our summaries independently, and then discussed them to arrive at the following consensus report.

Myelination is essential to brain health and function, but yet the cellular and molecular mechanisms that facilitate the wrapping and growth of myelin membrane on axons remain incompletely described. With this manuscript, the authors present data that they interpret as evidence that Vamp2/3-mediated exocytosis helps drive myelin membrane expansion on axons, at least in part by delivering components of myelin membrane to sites of membrane growth. If sufficiently well supported, this manuscript would likely stand as a significant contribution to the myelin and broader neuroscience fields, and stimulate new areas of important research.

The manuscript is carefully written, and conveys a clear and easy-to-follow message throughout. The experiments follow a fairly logical progression, and the experimental design, including validations, data acquisition, and data analyses appear to be sufficiently rigorous. The figures are nicely illustrated, organized, and annotated, and interpretation by a reader is straightforward. The Methods section is highly detailed and covers all features of experimental design. Altogether, the interpretations and claims made by the authors are strongly supported by the data.

As a very minor suggestion, the authors might consider broadening the possible interpretations of their data in the Discussion. Although their data nicely support the idea of vesicle-mediated membrane expansion, they do not exclude other possibilities. It could well be that disrupting exocytosis also disrupts, perhaps very indirectly, cell signaling mechanisms that also contribute to sheath growth. A brief "limitations of this study and alternative interpretations" section might be a useful way to help others think about building on this important study.

In summary, this manuscript addresses an important problem in neurobiology with a well-crafted experimental design and robust data. It will likely be quite interesting to the myelin community, to the wider neuroscience field, and to cell biologists.

Congratulations to the authors on a beautiful and significant contribution.

Bruce Appel

Reviewer #2 (Remarks to the Author):

This interesting study investigated the role of SNARE proteins VAMP2 and 3 in control of myelin formation by oligodendrocytes and Schwann cells. The expansion of membrane area and proper insertion of integral membrane proteins to form nodal domains are crucial events involved in myelination, but the molecular mechanisms that orchestrate the spatial and temporal dynamics of these events remain to be determined. The authors primarily make use of two manipulations in this study – conditional expression of botulinum neurotoxin B light chain in oligodendrocyte lineage cells using Cnp-Cre mice and imaging of VAMP-pHluorin constructs – which they use in combination with numerous assays to assess myelin formation, myelin structure, cell area, and surface protein expression. The authors find that VAMP cleavage in oligodendroglia in vivo results in hypomyelination, reduced formation of nodes of Ranvier, and premature death of the mice. Dynamic imaging of overexpressed VAMP-pHluorin in culture and in zebrafish in vivo, show that fusion of VAMP containing vesicles occurs primarily in oligodendrocyte processes and is enhanced as the cells transition from the progenitor to premyelinating stage. As expected, inhibition of VAMP-mediated exocytosis reduced surface expression of various proteins involved in myelination, as assessed

through surface biotinylation.

The manuscript is very clearly written and the experiments and analysis have been performed carefully. Together, these studies indicate that these SNARE proteins are required for oligodendrocytes to achieve their full size and establish normal myelin sheaths. There are several pieces of data which would enhance the impact of the studies and I have highlighted below instances where overarching statements and over interpretation should be revised to consider alternative explanations for the phenotypes observed. A major limitation of the studies is the use of Cnp-Cre mice, which appears to show variable penetrance and onset of recombination within the oligo lineage, and is complicated by the fact that this mouse line exhibits recombination in some neurons. Because these mice die prematurely (which is likely due to neuronal botox expression), it was not possible to assess the phenotype of oligodendrocytes and myelin in the adult CNS. The authors speculate about the possible regulation of VAMP2/3 in remyelination and adaptive myelination/plasticity, but don't perform a conditional expression of botox in mature oligodendrocytes to assess this possibility.

Main comments

1. The authors provide a description of the phenotype of myelin sheaths in the spinal cord of young postnatal mice (P12). If possible, it would be helpful to have a comparison of g-ratios between the genotypes.
2. The authors use Cnp-Cre mice to induce expression of botulinum neurotoxin B light chain in the oligodendrocyte lineage and perform quantification using GFP to assess the extent of expression. However, they neglect to describe recombination outside the oligodendrocyte lineage. There have been reports of Cnp-Cre expression among various neuron subtypes, which may have contributed to the reduced size and early lethality of these animals. The GRP+ motor neurons visible in the spinal cord sections are notable in this regard (Fig. 1c). The authors should formally address whether any of the myelin phenotypes could have been caused by expression outside the oligodendrocyte lineage.
3. Due to the early lethality of these mice, a major limitation of these studies is the inability to assess the phenotype of oligodendrocytes in the mature CNS. The in vitro studies (e.g. Fig. 3b, Ext. data 6a) reveal that older oligodendrocyte can achieve morphologies that approach those of WT. In this regard, although not essential for this study, it would be interesting to use PDGFRaCreER or NG2-CreER mice to express Botox, to ensure that VAMP cleavage occurs prior to differentiation. This would also offer the ability to titrate the extent of cell manipulation to avoid potential lethality and assist in evaluation of cell autonomous phenotypes. As the phenotype is partial – that is, oligodendrocytes are still able to expand their size and form some sheaths – the authors should discuss how this may occur in the absence of VAMP2/3 mediated fusion.
4. The use of the analysis metric “MBP coverage” is somewhat confusing. Does this refer to the intensity of fluorescence or area of fluorescence? The statement that there is reduced white matter volume is not readily apparent from the images provided. The area looks similar, but the intensity of the immunoreactivity is lower.
5. The authors note that “iBot;Cnp-Cre mice were severely hypomyelinated, with a 5-fold decrease in the number of wrapped axons and a concomitant 2.8-fold increase in the number of unmyelinated axons.” Shouldn't the decrease in myelinated axons be matched by an increase in unmyelinated axons if there is no neurodegeneration/axon atrophy?
6. The authors use VAMP-pHfluorin constructs to monitor vesicle exocytosis. This is a wonderful series of experiments, but there are several caveats that should be mentioned. Perhaps the most important is that both the in vitro and in vivo zebrafish experiments require overexpression of VAMPs. It is important for the authors to consider whether this manipulation may alter the frequency or location of fusion events. Because oligodendrocytes form flat sheaths in vitro is difficult to assess the significance of fusion in the processes, as the “processes” account for most of the membrane. Perhaps the best assessment of location comes from the zebrafish, but here it is not apparent from the images shown that fusion occurs at sites of nascent sheath formation/elongation. It would be helpful for the author temper their statements about sites of membrane insertion or provide additional evidence that fusion occurs along growing sheaths. It isn't clear why VAMP2 and VAMP3 would have different sites of fusion, as the authors suggest that they can compensate for one another.
7. The authors note that “iBot;Cnp-Cre oligodendrocytes exhibited a modest reduction in the number of sheaths per cell body compared to controls (Fig. 4c).” Although there is a trend, this effect was not statistically significant and so should not be reported as a reduction. Related, the statement that the “inability of these oligodendrocytes to elongate along axons leads to pronounced hypomyelination,”

seems an over interpretation of the data provided. As there are no time lapse movies following the formation of sheaths, it isn't clear how the cells arrive at the phenotypes observed.

8. Previous studies have examined the effects of tetanus toxin expression in zebrafish oligodendrocytes. It would be helpful to describe the phenotype of these mice in relation to VAMP function and whether there is potential for redundancy with VAMP3. In addition, it would be very interesting to assess the effects of botulinum neurotoxin B light chain expression by oligodendrocytes/premyelinating oligos in this system, as it would allow for dynamic analysis and assessments of mature phenotypes. If VAMP2/3 are dispensable for CNS myelinating in zebrafish, as shown for Schwann cells, then analysis of VAMP-pHluorins seems less relevant.

9. The description of the phenotypes observed in the TEM images of "vesicle accumulation" could also be interpreted as vacuolization. Moreover, the accumulation of vesicles reported in the high resolution light images look more like membrane blebbing (Ext. Fig 8). Similar phenotypes have been described in mice that carry mutations in connexins and ion channels. Therefore, it is possible that this could reflect impaired ion homeostasis by oligodendrocytes due to inability to deliver these membrane proteins. Some discussion of this possibility is warranted.

Minor comments

1. The authors note that "other VAMP2/3-dependent hits included intracellular membrane-proximal proteins such as MBP and ankyrin-G (AnkG)". It isn't clear why these intracellular proteins would be biotinylated during a surface labeling experiment.

2. A crucial part of the comparison involved assessments of membrane area. From the methods, this appears to have been estimated by measuring the cell border. As the cells are highly non-uniform and reticulated, this doesn't seem like area measurements based on cell border would provide a very accurate measure of membrane area.

3. Please separate the channels in Ext. Fig. 1d for Vamp2/3 for clarity.

4. "We next determined how myelin ultrastructure was affected using electron microscopy of the dorsal white matter of the thoracic spinal cord, which is comprised of parallel axon tracts" - as written it suggests you are looking at how EM affects the ultrastructure of myelin...

5. Prior studies of "kiss and run" fusion in neurons indicate that this often occurs repetitively before resulting in full fusion events. Do you see repetitive events at the same sites?

6. Reference formatting – "be VAMP2/3-independent (Song 2021)"

Reviewer #3 (Remarks to the Author):

This manuscript by Lam et al., in the Zuchero lab demonstrates a critical role for VAMP2/3- mediated exocytosis in membrane expansion of oligodendrocytes and consequentially the myelination of axons. Using an elegant combination of in vitro culture systems and clever genetic models to block VAMP2/3 exocytosis, the study demonstrates that VAMP2 and VAMP3 mediated exocytosis occurs primarily in the myelin sheaths paranodes, as oligodendrocytes mature over time, and are necessary for maturation. In vivo investigation of exocytosis in zebrafish supports conclusion. They go on to show in vitro and in vivo that vamp2/3 exocytosis is required for appropriate myelination and wrapping of axons with myelin and that in their absence, vesicles accumulate at the interface of myelin and axon, suggesting exocytosis at this inner tongue is required. Indeed Quantitative proteomics reveals reductions in the surface availability of myelin adhesions protein in the absence of VAMP2/3 mediated exocytosis suggesting exocytosis delivers such adhesion molecules necessary for the interface. This could very much explain why nodes of Ranvier fail to form in the absence of VAMP2/3 mediated exocytosis specifically in oligodendrocytes, and the subsequent failure to thrive and perinatal lethality. Overall, this is one of the strongest manuscripts I have reviewed with a stunning combination of in vitro and in vivo cell biology, beautiful imaging, rigorous quantification, intriguing proteomics that provide an interesting and relevant mechanisms, an exciting discussion that makes me excited for the next papers from the lab. Kudos to the authors, I have no additional experiments to suggest. Below I offer only minor suggestions for analysis, interpretation, and presentation.

1. for Fig 1fg, I might have expected more sheathed and not wrapped axons in the absence of exocytosis, if vamp2/3 is only needed for wrapping. However, how do you interpret that there are starting the process all together too (more nonmyelinated), if you presume vamp2/3 needed for going to sheathed wrapping.

As I was reading, I wondered why is there not a defect in PNS myelination, but the discussion of this difference and incorporation of lipoproteins was fascinating.

For movies, particularly Movie 1: I would suggest to equalize fluorescence (histogram matching) through movie so dimming overtime doesn't occlude visualization of events.

Figure 4: The difference of oligo cell shape and number of sheaths demonstrate in figure 4b between genotypes is quite striking. I almost wonder if the difference is underestimated, as it looks quite difficult to resolve number of sheaths in the wildtype cell. Could authors provide an example of an analyzed image in the extended data to show the number of sheaths/cell? Or potentially how an example in the control with less cells close together. Also I would imagine that analysis of cells closer to the aggregate versus further away (where axons are less dense) could also change the quantification significantly. Was this controlled for in the analysis?

In Figure 4F: are myelin sheaths/axons less organized, or organized differently, or are these distinct regions/ why so little gfp in top panel of f? maybe more comparable images would help?

Regarding vesicle accumulation in the absence of VAMP2/3 mediated exocytosis: In Extended data 8b is described as "accumulated vesicles were laterally distributed along the myelin sheath, rather than at the sheath edges." While I agree that the control and iBot:Cnp-Cre look very different, I don't agree or understand that description, and need clarification, and potentially quantification.

5B superresolution microscopy to investigate myelin in vivo is lovely and clearly demonstrates difference. , although images included probably need to be presented at higher resolution, could only resolve the sheaths in the inset. For quantification, variability of sheath may also be interesting to show (changes in varicosities etc). For reading the methods, it's not entirely clear if your method of analysis would fully capture this.

For figure 5c and movie, please include wildtype for comparison.

Lam, Takeo et al., Responses to Reviewers

The following regards the Reviewer comments made to our original submission, entitled “CNS myelination requires VAMP2/3-mediated membrane expansion in oligodendrocytes” (NCOMMS-22-07650-T).

We thank the reviewers for their helpful suggestions to enhance the manuscript, and for their recognition that our study “addresses an important problem in neurobiology” and is “a beautiful and significant contribution” (Reviewer #1), is “very clearly written and ... performed carefully” (Reviewer #2), and is “one of the strongest manuscripts I have reviewed” with “a stunning combination of in vitro and in vivo cell biology... that provide an interesting and relevant mechanism” (Reviewer #3). Both Reviewer #1 and Reviewer #3 had no major critiques of the work, while Reviewer #2 had several excellent points that we have now fully addressed.

To summarize our major revisions, we have added a new section titled “Limitations of this study and alternative interpretations”, as suggested by Reviewer #1 “to help others think about building on this important study”. We have also performed new experiments and discussed other studies that orthogonally “address whether any of the myelin phenotypes could have been caused by expression outside the oligodendrocyte lineage”, as suggested by Reviewer #2. Furthermore, we have revised the text and added new illustrations to enhance the clarity of our analysis and expand our interpretation of the results, as suggested by Reviewers #2 and 3.

Below we provide a point-by-point response to the reviewer comments (our responses in blue).

REVIEWER COMMENTS

Reviewer #1 (Remarks to the Author):

This review was completed with a graduate student as a training exercise. We read the manuscript and prepared our summaries independently, and then discussed them to arrive at the following consensus report.

Myelination is essential to brain health and function, but yet the cellular and molecular mechanisms that facilitate the wrapping and growth of myelin membrane on axons remain incompletely described. With this manuscript, the authors present data that they interpret as evidence that Vamp2/3-mediated exocytosis helps drive myelin membrane expansion on axons, at least in part by delivering components of myelin membrane to sites of membrane growth. If sufficiently well supported, this manuscript would likely stand as a significant contribution to the myelin and broader neuroscience fields, and stimulate new areas of important research.

The manuscript is carefully written, and conveys a clear and easy-to-follow message throughout. The experiments follow a fairly logical progression, and the experimental design, including validations, data acquisition, and data analyses appear to be sufficiently rigorous. The figures are nicely illustrated, organized, and annotated, and interpretation by a reader is straightforward. The Methods section is highly detailed and covers all features of experimental design. Altogether, the interpretations and claims made by the authors are strongly supported by the data.

As a very minor suggestion, the authors might consider broadening the possible interpretations of their data in the Discussion. Although their data nicely support the idea of vesicle-mediated

membrane expansion, they do not exclude other possibilities. It could well be that disrupting exocytosis also disrupts, perhaps very indirectly, cell signaling mechanisms that also contribute to sheath growth. A brief “limitations of this study and alternative interpretations” section might be a useful way to help others think about building on this important study. In summary, this manuscript addresses an important problem in neurobiology with a well-crafted experimental design and robust data. It will likely be quite interesting to the myelin community, to the wider neuroscience field, and to cell biologists.

Congratulations to the authors on a beautiful and significant contribution.

Bruce Appel

We thank the Reviewer for this generous and extremely positive review. In response to the suggestion above, we added a “**Limitations of this study and alternative interpretations**” section after our Discussion (starting on line ~425). Here we now consider additional indirect mechanisms that may also contribute the role of exocytosis in myelin sheath growth (line ~443), as well as several additional discussion points raised by Reviewers #2 and #3.

Reviewer #2 (Remarks to the Author):

This interesting study investigated the role of SNARE proteins VAMP2 and 3 in control of myelin formation by oligodendrocytes and Schwann cells. The expansion of membrane area and proper insertion of integral membrane proteins to form nodal domains are crucial events involved in myelination, but the molecular mechanisms that orchestrate the spatial and temporal dynamics of these events remain to be determined. The authors primarily make use of two manipulations in this study – conditional expression of botulinum neurotoxin B light chain in oligodendrocyte lineage cells using Cnp-Cre mice and imaging of VAMP-pHluorin constructs – which they use in combination with numerous assays to assess myelin formation, myelin structure, cell area, and surface protein expression. The authors find that VAMP cleavage in oligodendroglia in vivo results in hypomyelination, reduced formation of nodes of Ranvier, and premature death of the mice. Dynamic imaging of overexpressed VAMP-pHluorin in culture and in zebrafish in vivo, show that fusion of VAMP containing vesicles occurs primarily in oligodendrocyte processes and is enhanced as the cells transition from the progenitor to premyelinating stage. As expected, inhibition of VAMP-mediated exocytosis reduced surface expression of various proteins involved in myelination, as assessed through surface biotinylation.

The manuscript is very clearly written and the experiments and analysis have been performed carefully. Together, these studies indicate that these SNARE proteins are required for oligodendrocytes to achieve their full size and establish normal myelin sheaths. There are several pieces of data which would enhance the impact of the studies and I have highlighted below instances where overarching statements and over interpretation should be revised to consider alternative explanations for the phenotypes observed. A major limitation of the studies is the use of Cnp-Cre mice, which appears to show variable penetrance and onset of recombination within the oligo lineage, and is complicated by the fact that this mouse line exhibits recombination in some neurons. Because these mice die prematurely (which is likely due to neuronal botox expression), it was not possible to assess the phenotype of oligodendrocytes and myelin in the adult CNS. The authors speculate about the possible regulation of VAMP2/3 in remyelination and adaptive myelination/plasticity, but don't perform a conditional expression of botox in mature oligodendrocytes to assess this possibility.

We thank the reviewer for the positive and constructive feedback. We were able to fully address all of their major and minor points with added data, textual changes, and additional discussion. We will respond to each specific main and minor points below in-line.

Main comments

1. The authors provide a description of the phenotype of myelin sheaths in the spinal cord of young postnatal mice (P12). If possible, it would be helpful to have a comparison of g-ratios between the genotypes.

Great suggestion. We quantified the *g*-ratio between the genotypes and found that iBot mice have a pronounced and significant increase in *g*-ratio (i.e., thinner myelin) as was suggested by our EM micrographs in the original manuscript. These plots are now in Supplementary Fig. 2k-l.

2. The authors use *Cnp-Cre* mice to induce expression of botulinum neurotoxin B light chain in the oligodendrocyte lineage and perform quantification using GFP to assess the extent of expression. However, they neglect to describe recombination outside the oligodendrocyte lineage. There have been reports of *Cnp-Cre* expression among various neuron subtypes, which may have contributed to the reduced size and early lethality of these animals. The GRP+ motor neurons visible in the spinal cord sections are notable in this regard (Fig. 1c). The authors should formally address whether any of the myelin phenotypes could have been caused by expression outside the oligodendrocyte lineage.

This is an important point, and we apologize for not explicitly acknowledging the potential that *Cnp-Cre* expresses outside of oligodendrocytes and Schwann cells. (We used *Cnp-Cre* because it is the best currently available Cre line for targeting pre-myelinating oligodendrocytes.) In our revised manuscript, we have addressed this concern in several ways:

- a. Quantification of iBot expression in neurons (NeuN+;GFP+) has been added to Supplementary Fig. 2c-d, and showed that only 2-3% of GFP+ cells appear to be neurons. To address the extent of iBot expression in the region with GRP+ motor neurons, we have provided a zoomed inset in Supplementary Fig. 2c. The majority of GFP+ signal in this region overlaps closely with Olig2+ cell bodies (i.e., oligodendrocyte lineage cells), not neurons, as indicated by the white arrows. Note in this subpanel that oligodendrocyte cell bodies (Olig2, magenta) are often in close proximity to neuron cell bodies (NeuN, blue), but the GFP signal (yellow) almost always colocalizes with Olig2 signal. We would also like to point out that we did not detect widespread axon degradation by TEM (Fig. 1f) or a substantial change in neurofilament 200 (NF200) staining (Fig. 1e). However, we agree with the reviewer that even a small degree of iBot expression in neurons could contribute to some of the phenotypes we observed, so we addressed this in several additional ways:
- b. We have included new data using a completely orthogonal approach to test the role of VAMP2/3 on myelin sheath length. We tested the effect of oligodendrocyte-specific expression of dominant negative VAMP (dn-VAMP) constructs driven by the MBP promoter using neonatal injection of AAVs, which is shown in Supplementary Fig. 9e-i. This promoter was previously shown to be fully specific for mature oligodendrocytes when injected in the CNS of neonatal mice (von Jonquieres G... Klugmann M, 2013, PMID: 23799030). Our data demonstrated that oligodendrocytes expressing dn-VAMP2 in the P12 mouse spinal cord exhibited shorter myelin sheaths than the GFP-caax control, corroborating the sheath length defect observed in iBot;*Cnp-Cre* animals and showing that this phenotype is oligodendrocyte-autonomous.
- c. Two separate Cre lines (NG2-Cre and PDGFR α -CreERT) have been used by complimentary studies currently under review to express iBot in oligodendrocyte lineage

cells, and both Cre lines result in similar hypomyelination that we saw with *Cnp*-Cre (see full description below in response to Point #3). While no Cre line is perfect, NG2-Cre and PDGFR α -CreERT have not been reported to express in neurons to our knowledge. We have discussed these two separate studies in lines ~433-441 (see personal communications below).

- d. Based on a suggestion from Reviewer 1, we now have also added a section to our Discussion to explicitly mention the limitations of our study. In this section (line ~425), we discuss the nonspecific nature of *Cnp*-Cre and the several different ways we addressed this concern.
- e. We agree that the gross phenotypes of the mice (size, early lethality) could potentially be explained by iBot expression outside of oligodendrocytes and Schwann cells, and have now explicitly acknowledged this (lines 435-436). However, we would like to point out that other mouse mutants that are unable to form CNS nodes of Ranvier, like our iBot;*Cnp*-Cre mice, also die in the first 2-3 postnatal weeks (Susuki K... Rasband MN, 2013, PMID: 23664614).

3. Due to the early lethality of these mice, a major limitation of these studies is the inability to assess the phenotype of oligodendrocytes in the mature CNS. The in vitro studies (e.g. Fig. 3b, Ext. data 6a) reveal that older oligodendrocyte can achieve morphologies that approach those of WT. In this regard, although not essential for this study, it would be interesting to use PDGFR α CreER or NG2-CreER mice to express Botox, to ensure that VAMP cleavage occurs prior to differentiation. This would also offer the ability to titrate the extent of cell manipulation to avoid potential lethality and assist in evaluation of cell autonomous phenotypes.

[REDACTED]

[FIGURE REDACTED]

[REDACTED]

...As the phenotype is partial – that is, oligodendrocytes are still able to expand their size and form some sheaths – the authors should discuss how this may occur in the absence of VAMP2/3 mediated fusion.

Great suggestion: we now discuss additional mechanisms for myelin membrane expansion in the Discussion (Lines 355-361) and “Limitations of this study and alternative interpretations” (Lines 449-460) section.

4. The use of the analysis metric “MBP coverage” is somewhat confusing. Does this refer to the intensity of fluorescence or area of fluorescence? The statement that there is reduced white matter volume is not readily apparent from the images provided. The area looks similar, but the intensity of the immunoreactivity is lower.

We agree that this term was confusing. We have addressed this point in the following ways:

- a. We redefine this term as “white matter area” throughout the text, defining it as the percent area of a tissue section with above-threshold MBP immunostaining.
- b. We changed all instances of “MBP coverage” in the Figures to “% of MBP+ area” (Fig. 1d, Supplementary Figs. 3-4).
- c. Quantification of white matter area does show a robust and significant decrease in area (Fig. 1d). To better illustrate the difference, we have now added an additional panel in Supplementary Fig. 2h to display single-channel images of the MBP staining for all five biological replicate pairs analyzed in Fig 1d. These single-channel images better illustrate the reduction of white matter volume in *iBot;Cnp-Cre* mice.
- d. We also measured the mean MBP fluorescence intensity in all of these images and found that this is unchanged in *iBot* mice (Supplementary Fig. 2i). While MBP immunostaining intensity does not correlate well with degree of myelination (due to difficulties with antibodies penetrating and staining compact myelin), we interpret the similarities in MBP staining intensity *in vivo* to primarily reflect the fact that *iBot* oligodendrocytes are still able to differentiate normally—see Supplementary Fig. 2f (CC1 staining *in vivo*) and Supplementary Fig. 6d (MBP staining in culture).

5. The authors note that “*iBot;Cnp-Cre* mice were severely hypomyelinated, with a 5-fold decrease in the number of wrapped axons and a concomitant 2.8-fold increase in the number of unmyelinated axons.” Shouldn’t the decrease in myelinated axons be matched by an increase in unmyelinated axons if there is no neurodegeneration/axon atrophy?

We apologize that this initial description did not account for the difference in the total number of axons per area between the control and *iBot;Cnp-Cre* samples. In fact, we consistently found an increase in the number of *iBot;Cnp-Cre* axons per region due to their smaller caliber relative to control samples (see Fig. 1f). We have replotted our quantification of unmyelinated, ensheathed, and wrapped/wrapping axons as percentages of the total axon count to clarify the comparison between control and *iBot;Cnp-Cre* (revised Fig. 1h). To summarize our results, the percent unmyelinated axons increase from ~61% to ~90%, while the percent myelinated axons dropped concomitantly from ~29% to ~3% (while the percent ensheathed remained approximately the same at ~7-10%). This is a much more logical way to present the data, so we thank the reviewer for their insight.

6. The authors use VAMP-pHluorin constructs to monitor vesicle exocytosis. This is a wonderful series of experiments, but there are several caveats that should be mentioned. Perhaps the most important is that both the *in vitro* and *in vivo* zebrafish experiments require overexpression of VAMPs. It is important for the authors to consider whether this manipulation may alter the frequency or location of fusion events.

We are glad that the Reviewer appreciates this set of experiments (which were technically very demanding to set up). Overexpression of Vamp-pHluorin proteins is the standard method used in the field, and potential alternative approaches are either non-specific (e.g. dye labeling) or technically unfeasible (e.g. genome insertion to label endogenous Vamp proteins). Vamp-pHluorin overexpression has been used successfully in a variety of cell types, including neurons, astrocytes, and insulin-secreting cells (Sankaranarayanan S, Ryan TA, 2000, PMID: 10783237; Singh P... Zorec R, 2014 PMID: 24807050; Obermuller S... Barg S, 2005, PMID: 16141231). In neurons, Tim Ryan’s lab showed in a seminal paper that overexpressed Vamp2-pHluorin (aka “synapto-pHluorin”) localizes correctly and “does not compromise the secretory physiology of the synapse” (PMID:10783237). Therefore, live imaging of Vamp-pHluorins is the best way to measure exocytosis in oligodendrocytes with a great deal of precedence. However, this is a great point that we should have directly addressed and discussed.

Here is how we addressed this point in the revised manuscript:

- a. We determined whether Vamp-pHluorin overexpression perturbs exocytosis in oligodendrocytes by measuring the correlation between pHluorin expression and the number of events, by plotting the number of events vs. the mean fluorescence of each cell. We now include a graph of the number of events vs. mean fluorescence (as a proxy of Vamp-pHluorin expression) to determine whether the extent of Vamp-pHluorin expression impacts the frequency of exocytosis (Supplementary Fig. 5c). We found no correlation between Vamp-pHluorin expression level and number of events, suggesting that there is no dose-dependent effect of Vamp-pHluorin expression on exocytosis.
- b. To address the possibility that Vamp-pHluorin overexpression may perturb endogenous exocytosis in oligodendrocytes, we used cell area as an orthogonal metric to determine whether Vamp-pHluorin overexpression perturbs exocytosis-mediated membrane expansion. We found that the cell areas of oligodendrocytes expressing Vamp-pHluorin were within the normal range of cell areas of wild-type oligodendrocytes (Supplementary Fig. 5d), arguing that our experimental levels of Vamp-pHluorin expression did not grossly perturb oligodendrocyte exocytosis.
- c. We acknowledged this potential caveat alongside these new results in our revised results section (lines 155-158). We wrote: “Consistent with prior work showing that overexpressed VAMP2-pHluorin in neurons localizes correctly and does not perturb exocytic function (PMID:10783237), neither the frequency of exocytic events nor oligodendrocyte cell area was affected by VAMP-pHluorin overexpression level (Supplementary Fig. 5c-d).”

...Because oligodendrocytes form flat sheaths in vitro is difficult to assess the significance of fusion in the processes, as the “processes” account for most of the membrane. Perhaps the best assessment of location comes from the zebrafish, but here it is not apparent from the images shown that fusion occurs at sites of nascent sheath formation/elongation. It would be helpful for the author temper their statements about sites of membrane insertion or provide additional evidence that fusion occurs along growing sheaths. It isn't clear why VAMP2 and VAMP3 would have different sites of fusion, as the authors suggest that they can compensate for one another.

We tempered our statements about the precise locations in sheaths where VAMP2 or VAMP3 fuse as follows:

- d. Results, line 129 (Title of section) and Title of Figure 2: Toned down claims by changing from “VAMP2/3-mediated exocytosis is enriched in OL processes with a bias towards myelin sheath paranodes” to “VAMP2/3-mediated exocytosis occurs preferentially in myelin sheaths,” which is fully supported by all of our live-imaging data, and consistent with the preferential accumulation of aberrant vesicular structures in myelin that we quantify later in Figure 5.
- e. Results, line 153-154 (cultured oligodendrocytes): Previous statement read, “...indicating that VAMP2 and VAMP3 preferentially mediate vesicle fusion away from the soma”. We tempered this statement and clarified what we meant with this revised statement: “...suggesting that vesicles marked by VAMP2 and VAMP3 traffic away from the soma prior to exocytosis out in myelin sheets.”
- f. Results, lines 167-169 (in vivo in fish): Previous statement read, “Intriguingly, these results suggest a bias of VAMP2-mediated exocytosis at paranodes, while VAMP3-mediated exocytosis occurs indiscriminately between internodes and paranodes”. We

agree with the reviewer that this point is unnecessary to make here and would be better served as part of a future study, so we have deleted this sentence. Instead we now say, “Resolving how VAMP2- and VAMP3-mediated exocytosis are spatially regulated in sheaths remains an interesting question for future studies.”

- g. Figure 2f: We revised the graph to change “paranode” to “sheath edge”, since we did not directly stain for paranodes in these experiments.
- h. Discussion, line 370. Similar to above, we changed “VAMP2 events were preferentially distributed at paranodes” to “VAMP2 events were preferentially distributed at sheath edges,” since we did not directly stain for paranodes in these experiments.

We also should point out that although we saw that VAMP2 preferentially fused in sheath edges in vivo, both VAMP2 and VAMP3 have overlapping sites of fusion throughout internodes. So if there is in fact compensation between VAMP2 and VAMP3, this overlap could explain it. Importantly, though, we do not claim that VAMP2 and VAMP3 are able to compensate for one another—we only bring this up as a possibility in our Introduction (“compensation by VAMP2 may have limited the ability of previous VAMP3 KO studies”). We hope to resolve these questions in future studies that are beyond the scope of this manuscript.

7. The authors note that “iBot;Cnp-Cre oligodendrocytes exhibited a modest reduction in the number of sheaths per cell body compared to controls (Fig. 4c).” Although there is a trend, this effect was not statistically significant and so should not be reported as a reduction.

Corrected (lines 215-217). Line now reads, “iBot;Cnp-Cre oligodendrocytes did not have a statistically significant difference in the number of sheaths per cell body compared to controls (Fig. 4c) but formed significantly shorter sheaths...”

Related, the statement that the “inability of these oligodendrocytes to elongate along axons leads to pronounced hypomyelination,” seems an over interpretation of the data provided. As there are no time lapse movies following the formation of sheaths, it isn’t clear how the cells arrive at the phenotypes observed.

This is a fair point: short sheaths that we see in iBot mice could in theory be caused via multiple “paths”, not just reduced elongation (e.g. sheaths could form normally then shorten with time). We changed the text to “blocking exocytosis...results in shorter sheaths that leads to pronounced hypomyelination.” (lines 245-246).

8. Previous studies have examined the effects of tetanus toxin expression in zebrafish oligodendrocytes. It would be helpful to describe the phenotype of these mice in relation to VAMP function and whether there is potential for redundancy with VAMP3. In addition, it would be very interesting to assess the effects of botulinum neurotoxin B light chain expression by oligodendrocytes/premyelinating oligos in this system, as it would allow for dynamic analysis and assessments of mature phenotypes. If VAMP2/3 are dispensable for CNS myelinating in zebrafish, as shown for Schwann cells, then analysis of VAMP-pHluorins seems less relevant.

We believe the reviewer is referring to three papers from 2015-2016 (Mensch S, Lyons DA, 2015, PMID: 25849985; Hines JH... Appel B, 2015, PMID: 25849987; and Koudelka S... Lyons DA, 2016, PMID: 27161502) from David Lyons’ and Bruce Appel’s labs in which tetanus toxin (TeNT) was used to block Vamp function in zebrafish neurons, as a way to determine the non-cell autonomous effects of synaptic vesicle release on myelination (TeNT has the same protein

targets as iBot: Vamp1, 2, and 3). While the Appel paper (Hines et al. 2015) and the second Lyons paper (Koudelka et al. 2016) specifically expressed TeNT in neurons, the majority of experiments in the Lyons' group initial paper globally expressed TeNT—meaning that TeNT was expressed not only in neurons but also other cell types including oligodendrocytes (Mensch et al. 2015). There was a single experiment performed where TeNT was expressed exclusively in oligodendrocytes (see Mensch 2015 Supplementary Figure 7). They reported that TeNT expression in differentiated oligodendrocytes (using the MBP promoter) did not affect sheath number per oligodendrocyte. They did not, however, extensively characterize these fish, and did not comment on sheath length or myelin ultrastructure. We know of no other papers in which TeNT was expressed in zebrafish oligodendrocytes (Rafael Almeida, 3rd author on our manuscript, was in David Lyons' lab and is a coauthor on two of these three papers, so knows this literature extensively). We agree that it would be interesting to follow up on these prior studies and definitively test whether zebrafish oligodendrocytes require Vamp2/3 function to the same extent as mouse oligodendrocytes. However, these experiments are technically difficult and beyond the scope of this current manuscript, and we prefer to keep our story focused on mouse. We have added a mention of these previous studies to our “Limitations...” section (lines 427-429).

9. The description of the phenotypes observed in the TEM images of “vesicle accumulation” could also be interpreted as vacuolization. Moreover, the accumulation of vesicles reported in the high resolution light images look more like membrane blebbing (Ext. Fig 8). Similar phenotypes have been described in mice that carry mutations in connexins and ion channels. Therefore, it is possible that this could reflect impaired ion homeostasis by oligodendrocytes due to inability to deliver these membrane proteins. Some discussion of this possibility is warranted.

This is a very interesting suggestion. The figure described is now labeled as Supplementary Fig. 10 in the revised manuscript. The reviewer is correct that we do not know precisely what the vesicular structures are that accumulate in iBot mice, only that accumulation of vesicles under the membrane is consistent with blocking exocytosis. Our interpretation that these represent accumulated vesicles that are unable to fuse with the plasma membrane is consistent with numerous previous studies that found vesicles within the “cytoplasmic channels” and inner tongue of actively-growing myelin—see for example the beautiful examples using high pressure freezing from Snaidero N... Simons M, 2014, PMID:24439382 (notably Figure S4D-E, which highlights the high prevalence of these vesicles within myelin during development).

We also note that the vesicular structures we observe within myelin sheaths in iBot mice are ultrastructurally distinct from vacuoles as seen in connexin/ion channel mutants—vacuoles are entirely electron-lucent (see Menichella DM... Paul DL, 2003, PMID:12843301 on Cx47; vacuoles in Figs 7, 8, & 9) whereas accumulated vesicular structures in iBot mice have electron density similar to the vesicles seen in the Snaidero et al. 2014 paper described above, or cytoplasm (see our Figure 5e insets). Finally, we did not identify any connexins or ion channels as VAMP2/3-dependent hits in our mass spectrometry, although these proteins may lack exposed lysines that readily undergo surface biotinylation.

To address this point, we:

- a. Revised our initial mention of these from calling them “vesicles” to “vesicular structures” (e.g. lines ~254, 261, 263, 444, 445). When describing the EM results we now call them “structures resembling enlarged vesicles” (line ~261).
- b. Added a callout to the Discussion where we mention other possibilities (line ~273)

- c. Added discussion of alternative possibilities to the “Limitations of this study...” section, as requested by the reviewed (lines 444-447)

Minor comments

1. The authors note that “other VAMP2/3-dependent hits included intracellular membrane-proximal proteins such as MBP and ankyrin-G (AnkG)”. It isn’t clear why these intracellular proteins would be biotinylated during a surface labeling experiment.

These intracellular proteins were likely not directly biotinylated, but were enriched through their interaction with surface-biotinylated proteins. Previous proteomic studies using surface biotinylation have also reported co-purifying intracellular complexes residing at the plasma membrane (Smolders K... Arckens L, 2015, PMID: 26047021; Hormann K... Bennett KL, 2016, PMID: 26699813; Li M... Chen Y, 2021, PMID: 34178975). We note that proper targeting of VAMP2/3-trafficked surface proteins is also critical for the assembly of intracellular complexes tethered to the myelin membrane.

2. A crucial part of the comparison involved assessments of membrane area. From the methods, this appears to have been estimated by measuring the cell border. As the cells are highly non-uniform and reticulated, this doesn’t seem like area measurements based on cell border would provide a very accurate measure of membrane area.

We were careful to exclude regions within a cell that did not show GalCer staining. In Fig. 3b, we excluded all white “holes” created by highly reticulated regions from the membrane area calculation. Supplementary Fig. 10a shows a close-up of the GalCer staining to demonstrate that the lipid marker does covers the majority of a circular cell outline, leaving some “holes” near the cell body. To better illustrate our quantification, we have now added Supplementary Fig. 6e to display the GalCer immunofluorescence and the detected membrane area outline side-by-side.

3. Please separate the channels in Ext. Fig. 1d for Vamp2/3 for clarity.

We have separated the channels for Vamp2 and Vamp3 in Supplementary Fig. 1d.

4. “We next determined how myelin ultrastructure was affected using electron microscopy of the dorsal white matter of the thoracic spinal cord, which is comprised of parallel axon tracts” - as written it suggests you are looking at how EM affects the ultrastructure of myelin...

We corrected the text (lines 113-114). It now reads, “We next determined how myelin ultrastructure was affected in iBot;Cnp-Cre mice, using electron microscopy...”

5. Prior studies of “kiss and run” fusion in neurons indicate that this often occurs repetitively before resulting in full fusion events. Do you see repetitive events at the same sites?

Indeed, studies of single-vesicle fusion events induced by action potentials at presynaptic terminals have reported repeated “kiss-and-run” events with vesicle recycling on the milliseconds-to-seconds timescale (Aravanis AM... Tsien RW, 2003, PMID: 12789339; Gandhi SP, Stevens CF, 2003, PMID: 12789331; Sankaranarayanan S, Ryan TA, 2000, PMID: 10783237). In unstimulated oligodendrocytes, we observed at most two “kiss-and-run” events occurring at the same site within 1 minute, and estimate these repetitive events occur

infrequently in ~10% of all “kiss-and-run” events analyzed. Examples of repetitive events are shown in the density maps of Supplementary Fig. 5a, in which yellow spots indicate sites where two events occurred. At this moment, we do not think repetitive “kiss-and-run” events are a significant feature of isolated oligodendrocytes.

6. Reference formatting – “be VAMP2/3-independent (Song 2021)”

Thank you for catching this oversight; we have updated the reference.

Reviewer #3 (Remarks to the Author):

This manuscript by Lam et al., in the Zuchero lab demonstrates a critical role for VAMP2/3-mediated exocytosis in membrane expansion of oligodendrocytes and consequentially the myelination of axons. Using an elegant combination of in vitro culture systems and clever genetic models to block VAMP2/3 exocytosis, the study demonstrates that VAMP2 and VAMP3 mediated exocytosis occurs primarily in the myelin sheaths paranodes, as oligodendrocytes mature over time, and are necessary for maturation. In vivo investigation of exocytosis in zebrafish supports conclusion. They go on to show in vitro and in vivo that vamp2/3 exocytosis is required for appropriate myelination and wrapping of axons with myelin and that in their absence, vesicles accumulate at the interface of myelin and axon, suggesting exocytosis at this inner tongue is required. Indeed Quantitative proteomics reveals reductions in the surface availability of myelin adhesions protein in the absence of VAMP2/3 mediated exocytosis suggesting exocytosis delivers such adhesion molecules necessary for the interface. This could very much explain why nodes of Ranvier fail to form in the absence of VAMP2/3 mediated exocytosis specifically in oligodendrocytes, and the subsequent failure to thrive and perinatal lethality. Overall, this is one of the strongest manuscripts I have reviewed with a stunning combination of in vitro and in vivo cell biology, beautiful imaging, rigorous quantification, intriguing proteomics that provide an interesting and relevant mechanisms, an exciting discussion that makes me excited for the next papers from the lab. Kudos to the authors, I have no additional experiments to suggest. Below I offer only minor suggestions for analysis, interpretation, and presentation.

We thank the Reviewer for their very generous compliments on our work. We addressed each of their minor points below in-line (we numbered each Reviewer point for clarity).

1. for Fig 1fg, I might have expected more sheathed and not wrapped axons in the absence of exocytosis, if vamp2/3 is only needed for wrapping. However, how do you interpret that there are starting the process all together too (more nonmyelinated), if you presume vamp2/3 needed for going to sheathed wrapping.

Thank you for thinking deeply about our model. Our TEM data is in fact consistent with normal ensheathment and inhibited elongation. To better illustrate this concept, we have revised Supplementary Fig. 13a to include a theoretical example of how an ultrathin TEM section would show a higher frequency of unmyelinated axons in *iBot;Cnp-Cre*. Additionally, we have calculated the expected reduction in myelin coverage from shortened internodes (sheath length data) vs. from the reduced number of wrapped axons (TEM data), and the relative myelin coverage estimations are in close agreement. These calculations are detailed in a section titled “Quantification of myelin coverage from IHC internode analysis and electron microscopy sections” in the Methods (Lines 932-983).

2. As I was reading, I wondered why is there not a defect in PNS myelination, but the discussion of this difference and incorporation of lipoproteins was fascinating.

Thank you. We added a callout to “see Discussion” here so that readers will know that this will be addressed (line 126).

3. For movies, particularly Movie 1: I would suggest to equalize fluorescence (histogram matching) through movie so dimming overtime doesn’t occlude visualization of events.

Thank you for the suggestion. We have now applied bleach correction to minimize the dimming over time of Supplementary Video 1.

4. Figure 4: The difference of oligo cell shape and number of sheaths demonstrate in figure 4b between genotypes is quite striking. I almost wonder if the difference is underestimated, as it looks quite difficult to resolve number of sheaths in the wildtype cell. Could authors provide an example of an analyzed image in the extended data to show the number of sheaths/cell? Or potentially how an example in the control with less cells close together.

We have added an example of the sheath number analysis in Supplementary Fig. 8. In regions where two cell bodies had overlapping sheaths, we recorded the number of distinguishable sheaths divided by two, which occurred in about 11% of our segmented myelin sheath bundles. We have clarified our analysis in the Methods section (lines 918-921) and added a statement to acknowledge that the total number of sheaths per control cell may be underestimated due to the limited resolution of clustered sheaths.

5. ...Also I would imagine that analysis of cells closer to the aggregate versus further away (where axons are less dense) could also change the quantification significantly. Was this controlled for in the analysis?

This is an excellent point. We have included quantification of the axon density in each co-culture image in Supplementary Fig. 8c. We found no significant difference in the range of axon densities between RGC aggregates used with control or iBot;*Cnp-Cre* oligodendrocytes.

6. In Figure4F: are myelin sheaths/axons less organized, or organized differently, or are these distinct regions/ why so little gfp in top panel of f? maybe more comparable images would help?

Yes, the myelin sheaths and axons are less organized in the deep cortical layers examined for both control and iBot;*Cnp-Cre* in Fig. 4F. We do not expect GFP expression in the control sample (top panel of f), because iBot-IRES-GFP would not be expressed without *Cnp-Cre*. We have now clarified this point in the Figure Legend for Fig. 4f.

7. Regarding vesicle accumulation in the absence of VAMP2/3 mediated exocytosis: In Extended data 8b is described as “accumulated vesicles were laterally distributed along the myelin sheath, rather than at the sheath edges.” While I agree that the control and iBot;*Cnp-Cre* look very different, I don’t agree or understand that description, and need clarification, and potentially quantification.

We agree, and have softened our language to not over-interpret our findings. First, in response to Reviewer #2, we now refer to these as “vesicular structures.” We now mention the presence

of vesicular structure accumulation in both monocultures and cocultures without making any claims about their precise distribution in myelin sheaths. In addition, we quantified accumulated vesicles in two orthogonal ways in vivo: (1) using MBP immunostaining and super-resolution microscope (Figure 5b-c) and ultrastructurally by EM (Figure 5d-e), which we believe to be the ideal method to resolve and quantify the frequency of these vesicles.

8. 5B superresolution microscopy to investigate myelin in vivo is lovely and clearly demonstrates difference. , although images included probably need to be presented at higher resolution, could only resolve the sheaths In the inset.

Thank you for pointing this out. We have increased the resolution upon export of the file.

9. For quantification, variability of sheath may also be interesting to show (changes in varicosities etc). For reading the methods, it's not entirely clear if your method of analysis would fully capture this.

Thank you for this suggestion. We have now included an illustrated workflow of our diameter analysis in Supplementary Fig. 11. Our analysis does not account for variability within a single sheath. However, as a proxy for sheath variability, we have plotted the average standard deviation of diameters per biological sample in Supplementary Fig. 11e. The data show that iBot;*Cnp-Cre* sheath diameters have greater standard deviation than the controls.

10. For figure 5c and movie, please include wildtype for comparison.

Thank you for this suggestion. We have now included the 3D-reconstruction of myelin segments from control and iBot;*Cnp-Cre* samples in Fig. 5b and Supplementary Video 4.

REVIEWER COMMENTS

Reviewer #1 (Remarks to the Author):

I thought the authors did a commendable job responding to all the prior comments. I have nothing to add to my previous assessment of the manuscript's quality and significance.

Bruce Appel

Reviewer #2 (Remarks to the Author):

The authors have sufficiently addressed the concerns raised in the prior review, by providing additional analysis and clarification/rewording of the text.

Reviewer #3 (Remarks to the Author):

The revised manuscript is improved and ready for publication in my opinion. My kudos to the authors.

Lam, Takeo et al., Responses to Reviewers

We thank all three reviewers for their positive assessment of our revised manuscript, entitled “CNS myelination requires VAMP2/3-mediated membrane expansion in oligodendrocytes” (NCOMMS-22-07650A). We are grateful for their help in improving the original submission.

REVIEWERS' COMMENTS

Reviewer #1 (Remarks to the Author):

I thought the authors did a commendable job responding to all the prior comments. I have nothing to add to my previous assessment of the manuscript's quality and significance.

Bruce Appel

Reviewer #2 (Remarks to the Author):

The authors have sufficiently addressed the concerns raised in the prior review, by providing additional analysis and clarification/rewording of the text.

Reviewer #3 (Remarks to the Author):

The revised manuscript is improved and ready for publication in my opinion. My kudos to the authors.